# An ecological framework to understand the efficacy of fecal microbiota transplantation

Yandong Xiao[1,2], Marco Tulio Angulo [3,4], Songyang Lao[1], Scott T. Weiss[2] & Yang-Yu Liu [2,5✉]

Human gut microbiota plays critical roles in physiology and disease. Our understanding of ecological principles that govern the dynamics and resilience of this highly complex ecosystem remains rudimentary. This knowledge gap becomes more problematic as new approaches to modifying this ecosystem, such as fecal microbiota transplantation (FMT), are being developed as therapeutic interventions. Here we present an ecological framework to understand the efficacy of FMT in treating conditions associated with a disrupted gut microbiota, using the recurrent *Clostridioides difficile* infection as a prototype disease. This framework predicts several key factors that determine the efficacy of FMT. Moreover, it offers an efficient algorithm for the rational design of personalized probiotic cocktails to decolonize pathogens. We analyze data from both preclinical mouse experiments and a clinical trial of FMT to validate our theoretical framework. The presented results significantly improve our understanding of the ecological principles of FMT and have a positive translational impact on the rational design of general microbiota-based therapeutics.

[1] College of System Engineering, National University of Defense Technology, 410073 Changsha, Hunan, China. [2] Channing Division of Network Medicine, Brigham and Women's Hospital and Harvard Medical School, Boston, MA 02115, USA. [3] Institute of Mathematics, Universidad Nacional Autónoma de México, 76230 Juriquilla, Mexico. [4] National Council for Science and Technology (CONACyT), 03940 Mexico City, Mexico. [5] Center for Cancer Systems Biology, Dana-Farber Cancer Institute, Boston, MA 02115, USA. ✉email: yyl@channing.harvard.edu

Rather than simple passengers in and on our bodies, commensal microorganisms play key roles in human physiology and disease[1]. Propelled by metagenomics and next-generation sequencing technologies, many scientific advances have been made through the work of large-scale, consortium-driven microbiome projects[2,3]. Despite these technical advances that help us acquire more accurate organismal compositions and functional profiles of the human microbiome[4], there are still many fundamental questions to be addressed at the systems level. After all, microbes form very complex and dynamic ecosystems, which can be altered by dietary changes, medical interventions, and many other factors[5–7]. The alterability of our microbiome offers a promising future for practical microbiome-based therapies[8,9], such as fecal microbiota transplantation (FMT)[10,11], but also raises serious safety concerns[12–15]. Indeed, due to its high complexity, untargeted interventions could shift our microbiome to an undesired state with unintended health consequences.

In this article, we aim for understanding the ecological principles of FMT. During FMT, fecal material from a carefully screened, healthy donor is introduced to a recipient through the lower gastrointestinal (GI) tract via colonoscope or enema[16,17]; or the upper GI tract via nasogastric tube[16,18]; or with a capsulized, oral frozen inoculum[19,20]. Both absolute and relative contraindications have been proposed for donor screening[16,21,22]. Absolute contraindications include the risk of infectious agent, GI comorbidities, etc., while relative contraindications include history of major GI surgery, metabolic syndrome, systemic autoimmunity, etc. Fecal microbiota transplantation has been successfully used in the treatment of recurrent *Clostridioides difficile* infection (rCDI)[10,19,23–27]. Numerous case reports and cohort studies have described the use of FMT in patients with inflammatory bowel disease[28–31]. Fecal microbiota transplantation has also been experimentally used to treat many other GI diseases such as irritable bowel syndrome[32–34] and allergic colitis[35], as well as a variety of challenging non-GI disorders such as autism[36], obesity[37], multiple sclerosis[38], hepatic encephalopathy[39], and Parkinson's disease[40]. Larger multicenter studies and standardized double-blinded randomized clinical trials are certainly needed to fully evaluate the efficacy of FMT in treating those diseases beyond rCDI.

Although FMT is increasingly being explored as a potential treatment to optimize microbiota composition and functionality[41], rCDI is so far the only disease that has the most robust clinical evidence supporting the use of FMT[10,19,23–27]. As an anaerobic gram-positive, spore-forming, toxin-producing bacillus, *C. difficile* is transmitted among humans through the fecal-oral route, and has emerged as a major enteric pathogen with worldwide distribution, greatly increasing morbidity and mortality in hospitalized patients[42]. In the United States, *C. difficile* is the most frequently reported nosocomial pathogen. A surveillance study in 2011 identified 453,000 cases of CDI and 29,000 deaths associated with CDI; approximately a quarter of those infections were community-acquired[43]. Hospital-acquired CDI quadruples the cost of hospitalizations and increases annual expenditures by approximately $1.5 billion in the United States[44]. In a healthy gut microbiota, *C. difficile* is typically unable to colonize the gut in the presence of hundreds of strains of bacteria that are normally present (Fig. 1a). However, after broad-spectrum antibiotic administration that disrupts the healthy community (Fig. 1b), *C. difficile* spores ingested from the environment are able to germinate and grow within the gut and produce potent toxins[45,46], rendering the development of CDI (Fig. 1c). Because *C. difficile* can form spores that are not killed by most antibiotics, and because the normal microbiota is diminished, the infection is often poorly responsive to standard

antibiotics such as vancomycin or metronidazole, and CDI recurs in 27% and 24% of subjects, respectively[47]. The risk of further CDI increases with each subsequent recurrence: 30% after the first recurrence and up to 60% following two recurrences[48,49]. By contrast, after hundreds of treatments in many independent institutions, FMT has been shown to cure ~80% of the most recalcitrant rCDI cases that had previously failed standard antibiotic therapy[23] (Fig. 1d).

We have only recently started to understand the molecular mechanisms of FMT in treating rCDI[50–56]. We still do not quite understand why ~20% of rCDI patients relapse after FMT[27,57,58]. Very little is known about the long-term effects of FMT from the ecological perspective[54]. There are many puzzles in this field that need systems-level understanding and ecological explanations. For example, what are the key ecological factors that determine the success of FMT in treating rCDI? Does FMT work equally well in treating primary and recurrent CDI? Does donor–recipient compatibility matter at all for FMT success? If yes, how can we choose the best donor for a given recipient? How can we design probiotic cocktails containing only the effective components of FMT? Will there be a generic probiotic cocktail (i.e., a magic bullet) that works for every patient? If not, how can we design personalized probiotic cocktails? Systematically addressing these issues requires ecological thinking. Given that the human gut microbiota operates as a complex ecosystem, community ecology provides powerful tools to understand the driving factors shaping microbial diversity, interspecies interactions, and community structure[59–64].

In this article, by combining community ecology theory and network science, we propose a theoretical framework to reveal the ecological principles of FMT, using rCDI as a prototype disease. First, we propose an ecological modeling framework to simulate the FMT process. This modeling framework enables us to predict several key factors that determine the efficacy of FMT. Moreover, it helps us develop an efficient algorithm for the rational design of probiotic cocktails to decolonize a pathogenic species. (Note that donors and recipients discussed in the FMT simulations just represent the hosts of different simulated microbial communities. They should not be confused with real human subjects in clinical studies.) Second, we analyze real data to test our theoretical predictions. We demonstrate the ubiquitous network effect in real microbial communities. Then, we compare the taxonomic diversity of pre-FMT microbiota of responders and non-responders in a clinical trial of FMT. Finally, we numerically demonstrate the effectivity of probiotic cocktails designed by our algorithm to decolonize *C. difficile* from a real microbial community. The presented results offer new insights on the ecological principles that govern the dynamics and resilience of human gut microbiota, holding a translation promise for the rational design of more powerful microbiota-targeted therapeutics.

## Results

**An ecological modeling framework.** The classical generalized Lotka–Volterra (GLV) model has been used in several ecological modeling works of host-associated microbial communities[65–67]. In this work, we also use it to simulate the FMT process (see "Methods"). In particular, we model the gut microbiota of different hosts as different local communities assembled from a global species pool (or metacommunity) of $N$ species with universal population dynamics following the GLV model. Different local communities (e.g., the gut microbiota of the four subjects in Fig. 2a) are modeled as different subsystems (subnetworks) of the global ecological network (shown in the center of Fig. 2a). This modeling procedure is inspired by the fact that different community assemblies could give rise to the highly personalized compositions observed in

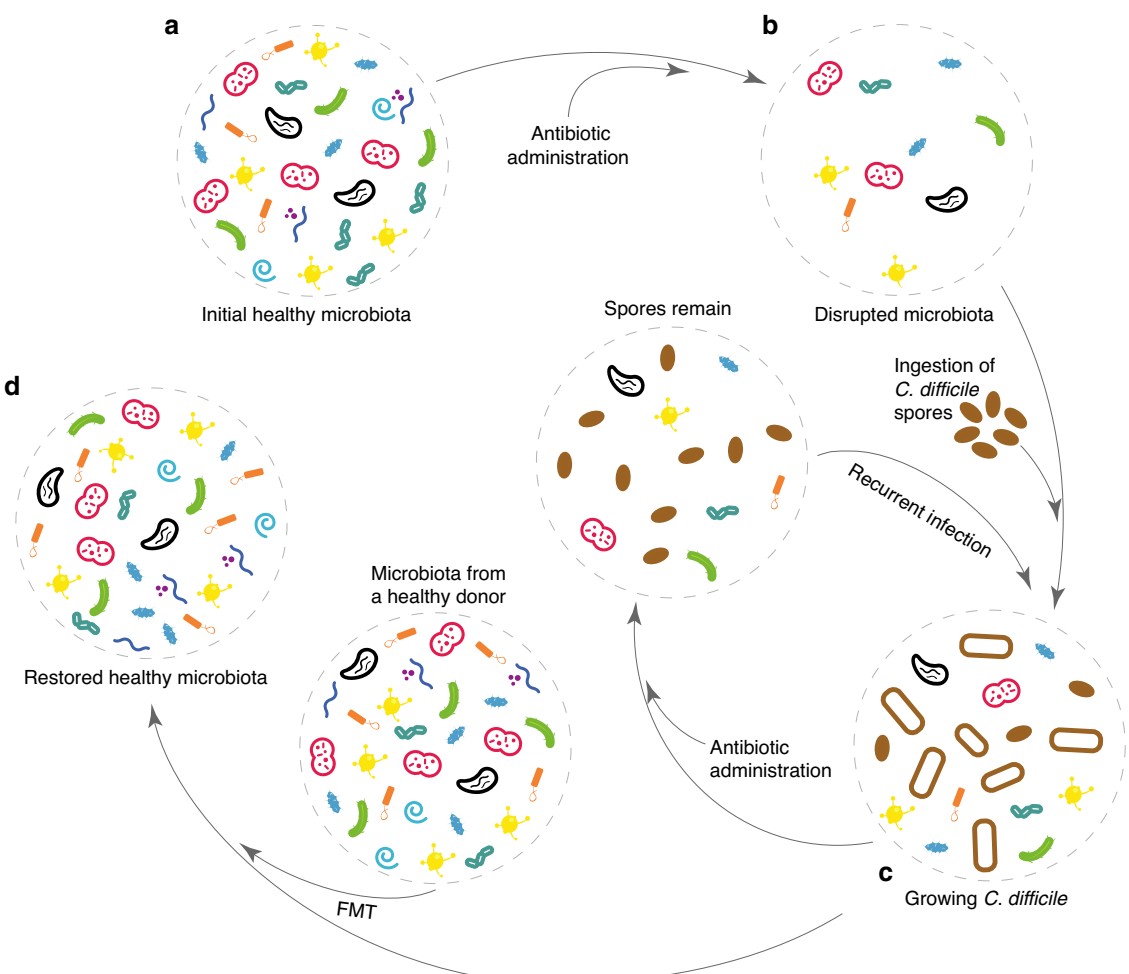

**Fig. 1 Fecal microbiota transplantation (FMT) aims to restore a healthy gut microbiota. a** In an initial healthy gut, the microbial community typically contains many different taxa and displays very high taxonomic and functional diversity. Most of those taxa are benign. Some of them can even keep out opportunistic pathogens such as *C. difficile*. **b** Antibiotic administration leads to low taxonomic diversity and to a disrupted gut microbiota, which allows colonization by *C. difficile*. **c** *C. difficile* spores are typically ingested following contact with contaminated biotic or abiotic surfaces, then germinate in the gut to a vegetative cell-type and produce potent gut-damaging toxins during a late growth stage. This leads the development of *C. difficile* infection (CDI). Ironically, standard treatment of CDI generally involves prescription of antibiotics such as metronidazole or vancomycin. Those antibiotics kill *C. difficile* but spores can remain in the gut, rendering recurrent CDI (rCDI). **d** Transplanting the fecal material from a healthy donor to the patient's gut can restore the healthy gut microbiota.

the human microbiome[2], as well as the recent finding that human gut microbiome displays strong universal dynamics for healthy adults[68].

To simulate the FMT process in treating rCDI, we consider a hypothetical ecological network (e.g., the central one shown in Fig. 2a), and numerically solve the corresponding GLV model for the time-dependent species abundances, which can be further visualized as time series (Fig. 2b). Time series data from different initial conditions can be visualized as different trajectories in the principal coordinate analysis (PCoA) plot (Fig. 2c). We use different species collections to simulate the donor's gut microbial composition, as well as the initial healthy state, the pre-FMT diseased state, and the post-FMT state of the recipient's gut microbial compositions. We assume that the healthy and diseased states of the gut microbiota can be coarsely characterized by the abundance of the *C* species, representing *C. difficile*, and our goal is simply to decolonize or suppress the growth of *C. difficile*.

To simulate the donor's healthy gut microbial composition, we randomly assemble a local community from the species pool with the only condition that *C. difficile* cannot colonize. (This is of course a simplified modeling approach. In reality, there are

asymptomatic carriers[69], i.e., with presence of toxicogenic *C. difficile* in their colon but no symptoms of CDI. This is not considered as healthy in our modeling framework.) To simulate the recipient's initial healthy state, we randomly assemble a local community from the species pool such that *C. difficile* abundance is very low (see Fig. 2d for the recipient's initial healthy ecological network). After broad-spectrum antibiotic administration, most of commensal species in the recipient's initial healthy microbiota are removed, including those that can inhibit the growth of *C. difficile*[70]; hence, *C. difficile* is able to grow, leading to the diseased state (Fig. 2e). (One can also assume that *C. difficile* is absent in the recipient's initial healthy microbiota, then simulate the ingestion of *C. difficile* spores after antibiotic administration, and the overgrowth of *C. difficile* in the recipient's gut microbiota. This process will yield very similar time series data and diseased state as our simulation does.) With FMT using fecal material from a healthy donor, we aim to decolonize *C. difficile* or at least reduce its abundance to the initial normal level by introducing back some species killed by antibiotics, and very likely some donor-specific species (e.g., species 5 in the recipient's post-FMT ecological network, Fig. 2f). The post-FMT community will

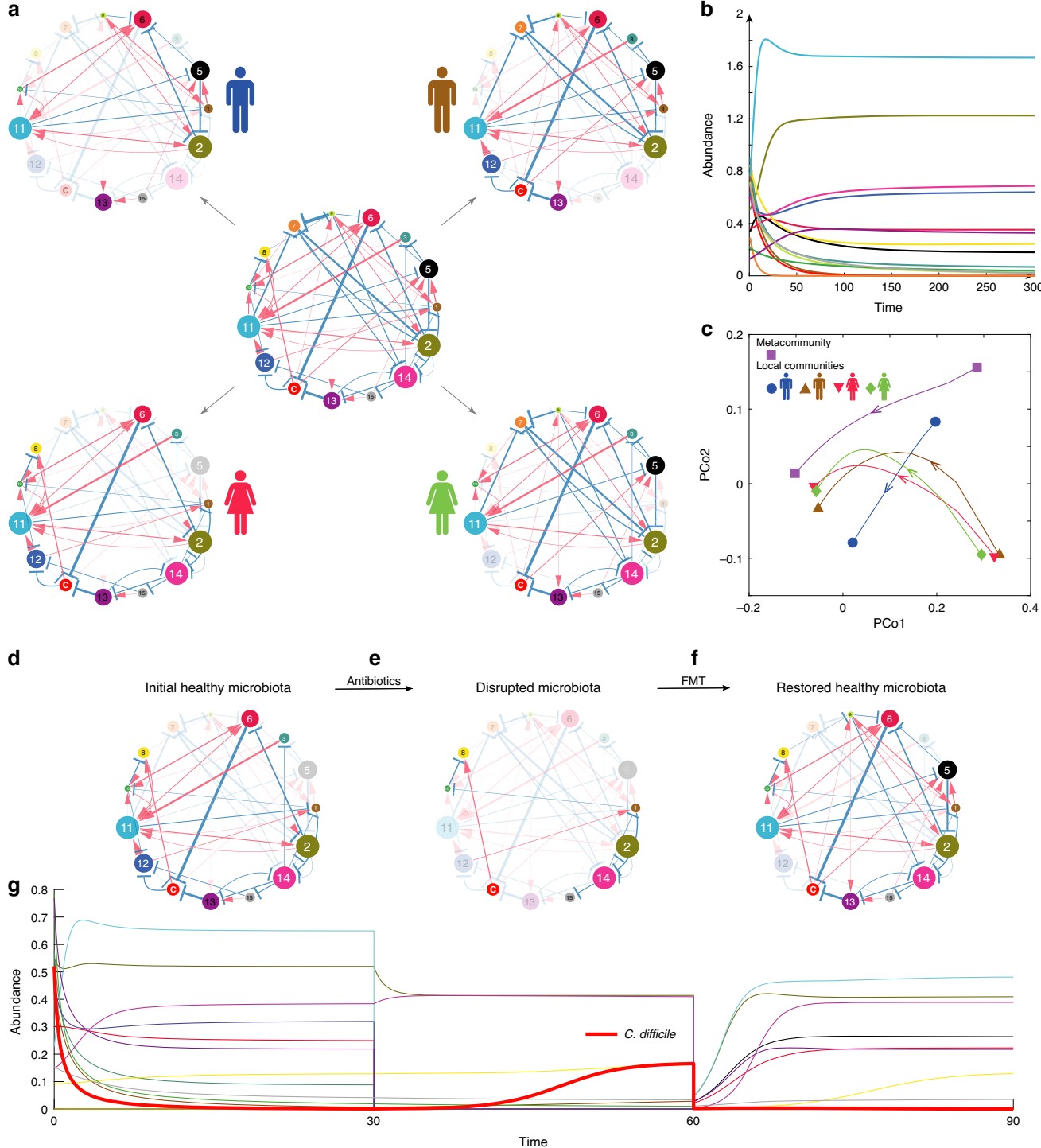

evolve into a new healthy state, corresponding to the recipient's restored healthy microbiota. The whole process can be visualized as time series of species abundances (Fig. 2g). See Supplementary Note 1 and Supplementary Figs. 1–3 for details of the FMT simulation process.

To quantify the efficacy of FMT in our simulations, we define a dimensionless variable called the recovery degree, $\eta = \frac{x^{(d)} - x^{(p)}}{x^{(d)} - x^{(h)}}$, where $x$ represents the abundance of *C. difficile*, and the superscripts (d), (p) and (h) represent the diseased, post-FMT, and initial healthy state of the recipient, respectively. For a successful FMT, the post-FMT *C. difficile* abundance $x^{(p)}$ should be similar to that of the initial healthy state $x^{(h)}$, rendering $\eta \approx 1$.

By contrast, if the post-FMT *C. difficile* abundance $x^{(p)}$ is quite similar to (or even higher than) that of the diseased state $x^{(d)}$, then $\eta \approx 0$ (<0), indicating a failure of FMT in our simulations.

**Impact of host-dependent microbial dynamics on FMT efficacy.** In a previous work[68], we raised a fundamental question on the universality of microbial dynamics: Are the ecological parameters (e.g., intrinsic growth rates, interspecies interactions) strongly host-dependent or largely host-independent (universal)? To illustrate the impact of host-dependent microbial dynamics on the efficacy of FMT, we perform extensive FMT simulations using microbial dynamics with four increasing levels of

**Fig. 2 An ecological modeling framework to study the dynamics of microbial communities. a** The underlying ecological network of a metacommunity with 15 different microbial species is a directed, signed and weighted graph, where blue/red edges represent negative/positive ecological interactions, and edge weights represent interaction strength. Different subjects can be considered as different local communities assembled from the species pool. Nodes in shadow denote species are not present in the local community. Then the ecological network associated with each local community (i.e., four subjects in panel (**a**)) is just a particular subgraph of the global ecological network associated with the metacommunity. **b** The temporal behavior of the metacommunity can be simulated by numerically solving the population dynamics model as shown in Eq. (1) for any given initial condition, i.e., the 15 species' initial abundances. The temporal behavior of each local community can be simulated similarly. **c** Principle Coordinate Analysis (PCoA) of the time series data of relative abundances calculated from both the metacommunity and the four different local communities. The root Jensen—Shannon divergence (rJSD) is used in the PCoA throughout this manuscript. **d** We start from a subject's initial healthy gut microbiota represented by an ecological network. Species "C" represents *C. difficile*. Note that the abundance of *C. difficile* is very low due to the inhibition of the other species. **e** A hypothetic broad-spectrum antibiotics eliminate many species from the initial healthy microbiota, rendering a disrupted microbiota represented by an ecological network with fewer nodes. **f** By transplanting fecal material of a healthy donor to the patient's gut, FMT can restore a healthy gut microbiota with high taxonomic diversity, represented by an ecological network with more nodes. Some of the transplanted species can effectively inhibit the growth of *C. difficile*, leading to a quick resolution of symptoms. **g** The simulated time series of species abundances. Thick red curve represents the time-dependent abundance of *C. difficile*. Note that around time ~60, the abundances of species in the pre-FMT microbiota drop drastically, because in our simulation we tried to mimic the pre-FMT bowel cleansing process in the clinical practice of FMT.

host-dependency based on the differences between the ecological networks of donor's and recipient's gut microbiota: Universal: the two networks are exactly the same (Fig. 3a); Host-Dependency-I: the two networks share the same structure and sign pattern (i.e., interaction types), but have different link weights (i.e., interaction strengths) (Fig. 3b); Host-Dependency-II: the two networks share the same structure, but different sign patterns (interaction types) and different link weights (interaction strengths) (Fig. 3c); Host-Dependency-III: the two networks have totally different structures, different sign patterns (interaction types), and different link weights (interaction strengths) (Fig. 3d).

In our FMT simulations, we consider a pool of 100 species for the metacommunity. The local communities corresponding to the donor's healthy state, the recipient's initial healthy state, and pre-FMT diseased state can be assembled as descried in Fig. 2. For each level of host-dependency of the microbial dynamics, in our FMT simulations we randomly choose 20 healthy donors to perform the FMT for a particular recipient, rendering 20 trajectories in each of the PCoA plots (Fig. 3e). Note that the healthy microbiota of those donors contains 60–80 species, while the recipient's pre-FMT diseased state contains only a small fraction (19/100) of species. The ecological networks of their gut microbiota are constructed based on the level of host-dependency. We assume that once the species from the donor's microbiota were transplanted to the recipient, these species follow the recipient's microbial dynamics. The colors of those FMT trajectories are based on the recovery degree $\eta$ evaluated after the recipient's gut microbiota reaches its steady state. We find that, with universal dynamics, FMT generally succeeds with very high recover degree for all the donors (i.e., $\eta \approx 1$). Yet, with increasing level of host-dependency in the microbial dynamics (i.e., the recipient's post-FMT gut microbiota becomes more different from that of the donor), more and more FMT processes will yield very a low recovery degree ($\eta \approx 0$ or <0), corresponding to those blue trajectories in Fig. 3e. In other words, more and more donors will fail in restoring the recipient's healthy gut microbiota.

To systematically study the impact of host-dependent microbial dynamics on the FMT efficacy, for each level of host-dependency, we now consider 50 donors and 50 recipients in our FMT simulations. The ecological networks of their gut microbiota are constructed based on the level of host-dependency. During each FMT simulation, we transplant the microbiota of a randomly chosen healthy donor to a randomly chosen recipient. We find that the higher the host-dependency level of the microbial dynamics, the lower the FMT efficacy (Fig. 3f). This theoretical result agrees well with our intuition, because

host-dependent microbial dynamics may drive the ecosystem to unexpected states with unintended consequences. For example, if interaction types are host-dependent, then the same set of species that inhibit the growth of *C. difficile* in the donor's gut might promote the growth of *C. difficile* in the recipient's gut.

Since the high efficacy of FMT for rCDI has been shown with robust clinical evidence[10,19,23–27], our theoretical result suggests that the existence of host-dependent microbial dynamics for rCDI patients and their donors is highly improbable. Moreover, it implies that universal microbial dynamics[68] (or at least very low level of host-dependency) is a key factor determining the success of FMT. Therefore, in the FMT simulations conducted in the following subsections, for the sake of simplicity, we assume all the local microbial communities have universal dynamics.

**Impact of pre-FMT taxonomic diversity on FMT efficacy.** Standard treatment of first episode of CDI generally involves prescription of either metronidazole or vancomycin[71]. There is insufficient evidence to recommend FMT as a treatment for the first episode of CDI, while recent clinical practice guidelines strongly recommend FMT as a first-line treatment option for both mild and severe rCDI with high quality of evidence[21,71,72]. It has been shown that primary CDI patients have less dysbiotic and more diverse gut microbiota than that of rCDI patient[73]. This prompts us to systematically study the impact of taxonomic diversity of the recipient's pre-FMT microbiota on the FMT efficacy using our ecological modeling framework.

We perform FMT simulations with a pool of 100 species for the metacommunity. Local communities corresponding to the donor's healthy gut microbiota and the recipient's diseased gut microbiota are assembled as described in the previous section. In our FMT simulations, we find that if the recipient's diseased state has very low species richness, then FMT will work with almost all the donors (Fig. 4a). By contrast, if the diseased state has higher species richness, then FMT will work with fewer donors (Fig. 4b, c), rendering lower FMT efficacy. Note that with increasing level of species richness in the diseased state, the FMT trajectory also looks more chaotic or irregular. We also plot the FMT efficacy (in terms of the recovery degree $\eta$) as a function of the taxonomic diversity (in terms of three different indices: species richness, Shannon entropy, and Simpson index) of the diseased state, finding that the FMT efficacy generally decreases with increasing taxonomic diversity of the diseased state (Fig. 4d–f).

The results demonstrated in Fig. 4 can be partially explained by the network effect: those species that either directly inhibit or have no direct impact on the growth of *C. difficile* might

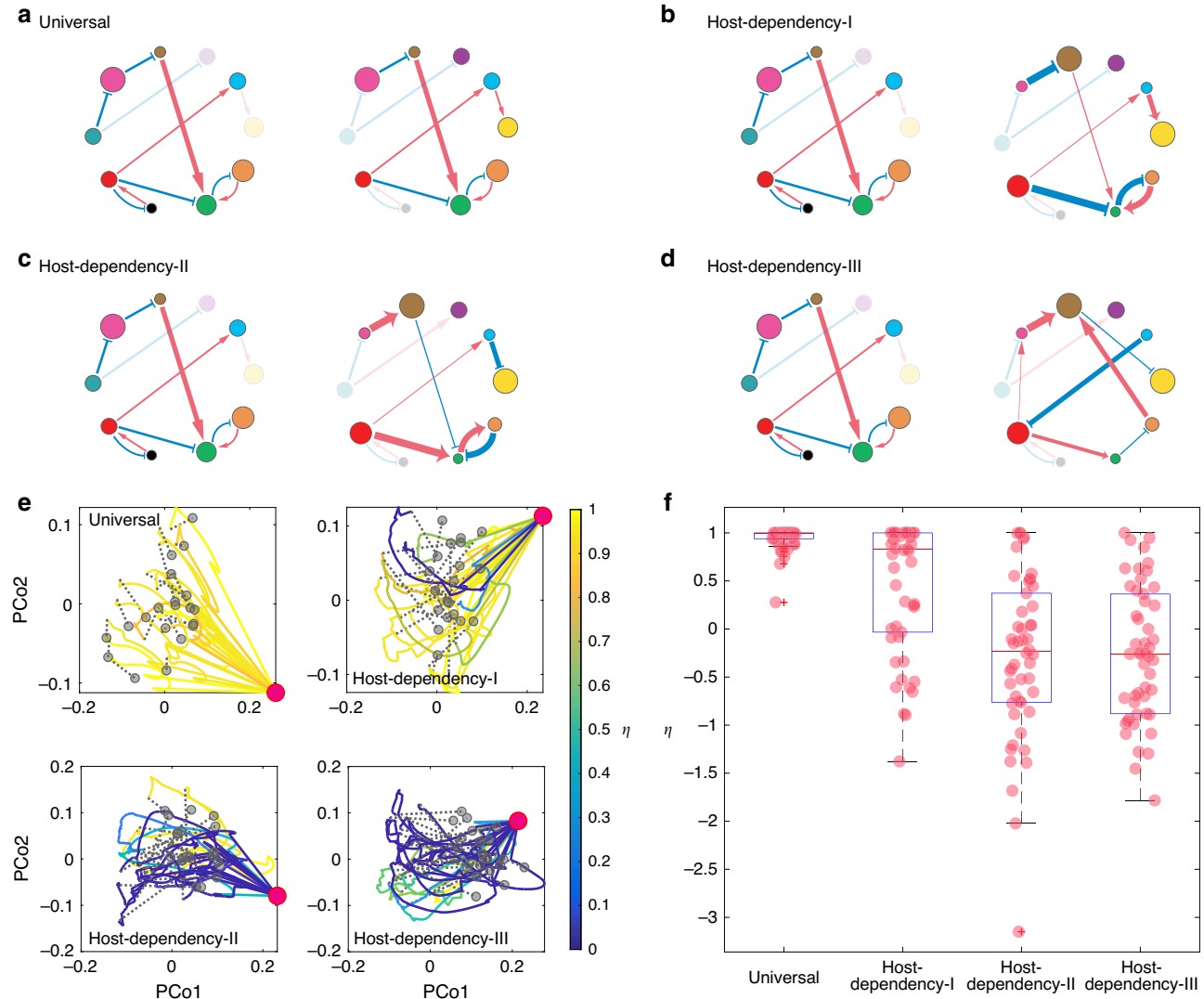

**Fig. 3 Impact of host-dependent microbial dynamics on FMT efficacy. a–d** Microbial dynamics with four increasing levels of host-dependency, visualized using the ecological network of the donor's (left) and recipient's (right) gut microbiota: **a** Universal: the two networks are exactly the same. **b** Host-Dependency-I: they share the same network structure and sign pattern (i.e., interaction types), but have different link weights (i.e., interaction strengths). **c** Host-Dependency-II: they share the same network structure, but different sign patterns and different link weights. **d** Host-Dependency-III: they have totally different network structures, different sign patterns, and different link weights. **e** We performed FMT simulations for each of the four cases (Universal, Host-Dependency-I, Host-Dependency-II, and Host-Dependency-III). In our simulations, we consider a global pool of 100 species, and a particular CDI patient with the diseased state containing only 19 of the 100 species (represented by the red dot in the PCoA plot). We randomly choose 20 healthy donors (represented by the gray dots in the PCoA plot) to perform the FMT for this patient and visualize the time series data of species abundances as trajectories in the PCoA plot, with trajectory color representing the recovery degree evaluated after the recipient's gut microbiota reaches steady state. Note that $\eta$ does not naturally belong to [0,1]. For the visualization purpose, we forced $\eta > 1$ as $\eta = 1$ and $\eta < 0$ as $\eta = 0$. **f** The recovery degree decreases with increasing level of host-dependency in the microbial dynamics. The model parameters used in the simulations of panels (**e**) and (**f**) are described in Supplementary Note 1. The boxplot is calculated from 50 independent FMT simulations. The central red line indicates the median; the bottom and top edges of the box indicate the 25th and 75th percentiles, respectively. The whiskers extend to the most extreme data points not considered outliers, and the outliers are plotted individually using the cross symbol.

indirectly promote the growth of *C. difficile* through other mediator species. The net or effective impact of a species on the growth of *C. difficile* is hence largely context-dependent (see Supplementary Note 2 and Supplementary Figs. 4 and 5 for detailed discussions and calculations of the net impact). If the recipient's diseased state has very low species richness, then the transplanted donor microbiota will not be highly influenced by those existing species. In particular, those species that effectively inhibit the growth of *C. difficile* in the donor's gut could still function similarly in the recipient's gut, rendering FMT with high efficacy. By contrast, if the recipient's diseased state has very high

species richness, then the transplanted donor microbiota will be strongly influenced by those existing species. Probably the donor microbiota cannot easily engraft because of competition between the recipient's existing species and the species in the FMT. Interestingly, as long as the species richness of donors' microbiota is above a certain threshold, it does not drastically affect the FMT efficacy (see Supplementary Fig. 10).

**Impact of donor−recipient compatibility on FMT efficacy.** For the clinical practice of FMT, detailed donor selection guidelines

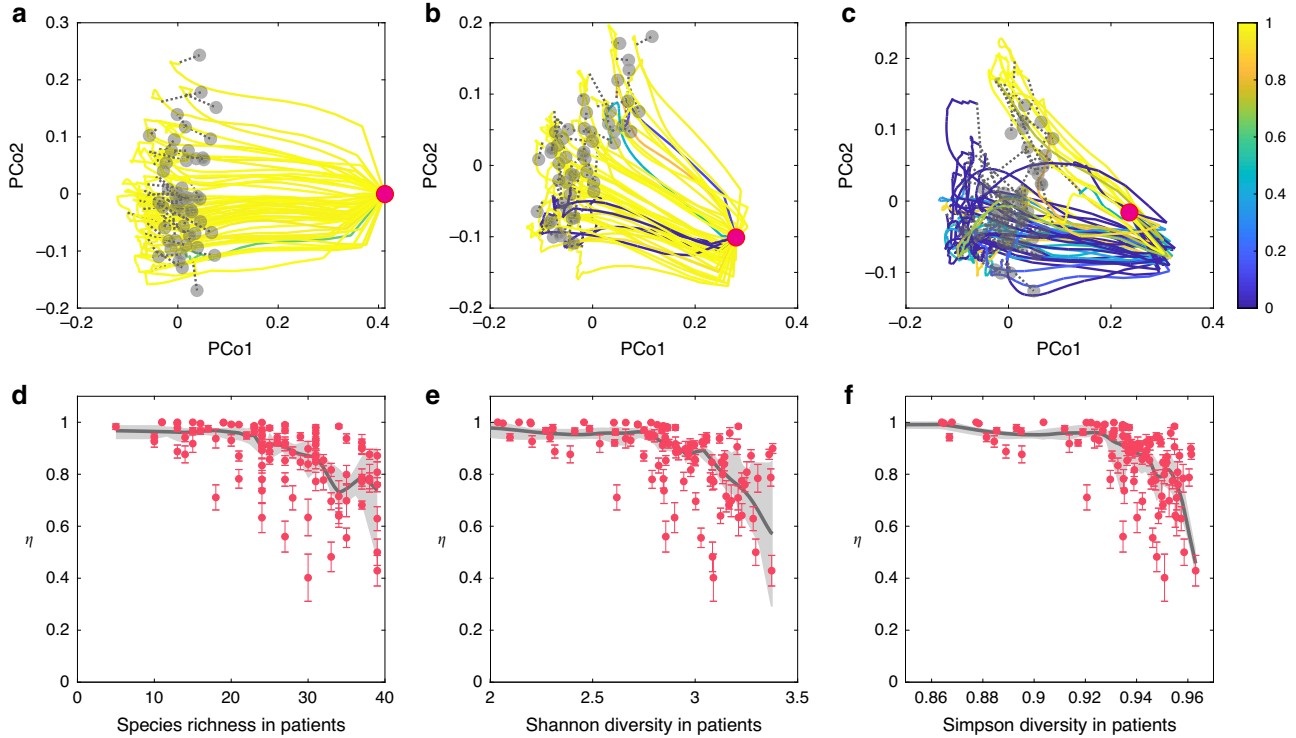

**Fig. 4 Impact of taxonomic diversity of recipient's pre-FMT microbiota on FMT efficacy.** We simulate the FMT processes with universal microbial dynamics. Species in the gut microbiota of 50 healthy donors are randomly chosen from a pool of 100 species with the only condition that *C. difficile* cannot colonize. The healthy microbiota of those donors (represented by the gray dots in the PCoA plot) contains 60−80 species. **a–c** The recipient's pre-FMT microbiota (represented by the red dot in the PCoA plot) has 12 (**a**), 22 (**b**), and 31 (**c**) species, respectively. For each case, we visualize the time series data of species abundances as trajectories in the PCoA plot, with trajectory color representing the recovery degree evaluated after the recipient's gut microbiota reaches steady state. **d–f** The taxonomic diversity of 100 recipients' pre-FMT microbiota (quantified by three indices: species richness (**d**), Shannon diversity (**e**), and Simpson diversity (**f**)) strong affects the FMT efficacy (quantified by the recovery degree). We performed nonparametric regression and bootstrap sampling to calculate the trend (black line) and its 94% confidence interval (gray shadow). The error bar represents the standard error of the mean (SEM) calculated from 50 independent FMT simulations. See Supplementary Note 1 for details on the model parameters used in the FMT simulations.

have been listed in an evidence-based report[21] and a recently published correspondence[22]. For inflammatory bowel disease, it has been shown that the donor's microbial diversity has an influential role in the therapeutic success of FMT[74,75]. For rCDI, previous studies have shown that the choice of donor, be it a relative, spouse, or anonymous volunteer, does not appear to influence the clinical efficacy of FMT in treating rCDI[23,76]. Currently, there are no clinical guidelines for matching a donor to a particular recipient. Our simulation results suggest that for rCDI patients with low taxonomic diversity, FMT should work equally well with different donors (Fig. 4a). Moreover, our simulations suggest that the FMT efficacy will decrease with increasing taxonomic diversity in rCDI patients (Fig. 4d–f), implying a pronounced donor−recipient compatibility issue.

To systematically study this intriguing compatibility issue, we performed extensive FMT simulations by considering three sets of 50 recipients with increasing level of taxonomic diversity, and a fixed set of 100 donors. For each (donor, recipient) pair, we perform an FMT simulation and calculate the recovery degree. The result is shown as heat maps in Fig. 5a–c. There are several interesting findings in our simulations. First, some donors work for all the recipients. Those donors can be considered to be super donors[77]. The chance to find a super donor becomes lower if recipients have higher taxonomic diversity in their pre-FMT microbiota. Second, some recipients can be treated with all the donors. Those recipients can be considered to be super recipients. With higher taxonomic diversity in the pre-FMT microbiota, it is

harder to identify those super recipients. Third, for each of those non-super donors, different recipients can have very different recovery degrees. This is consistent with the previous finding that with the same donor different recipients can have very different post-FMT microbiota[78]. Finally, for recipients with higher taxonomic diversity in their pre-FMT microbiota, a larger fraction of donors will not work, consistent with our results shown in Fig. 4.

To find a compatible donor for a given recipient in the clinical practice of FMT, one may consider the following naive approach: compare their microbial compositions, and calculate the fractions of donor-specific taxa (denoted as $f_d$), recipient-specific taxa ($f_r$), and common taxa ($f_c = 1 - f_d - f_r$). Based on the simulation results shown in Fig. 5a–c, we investigated the relationship between FMT efficacy and the three fractions ($f_d, f_r, f_c$). The result is shown as ternary plots in Fig. 5d–f. Interestingly, compatible and noncompatible pairs are well mixed, implying that it is impossible to distinguish them solely based on ($f_d, f_r, f_c$). Note that in a previous work[79], it was found that donor microbiota engraftment can be predicted largely from the abundance and phylogeny of bacteria in the donor and the pre-FMT microbiota of the recipient. This does not contradict our simulation results presented in Fig. 5d–f because here we care more about FMT efficacy in decolonizing the pathogenic species and hence treating rCDI. Donor microbiota engraftment is a necessary condition for FMT success, but it is not sufficient. Colonized taxa have to be able to effectively inhibit the growth of *C. difficile* as well.

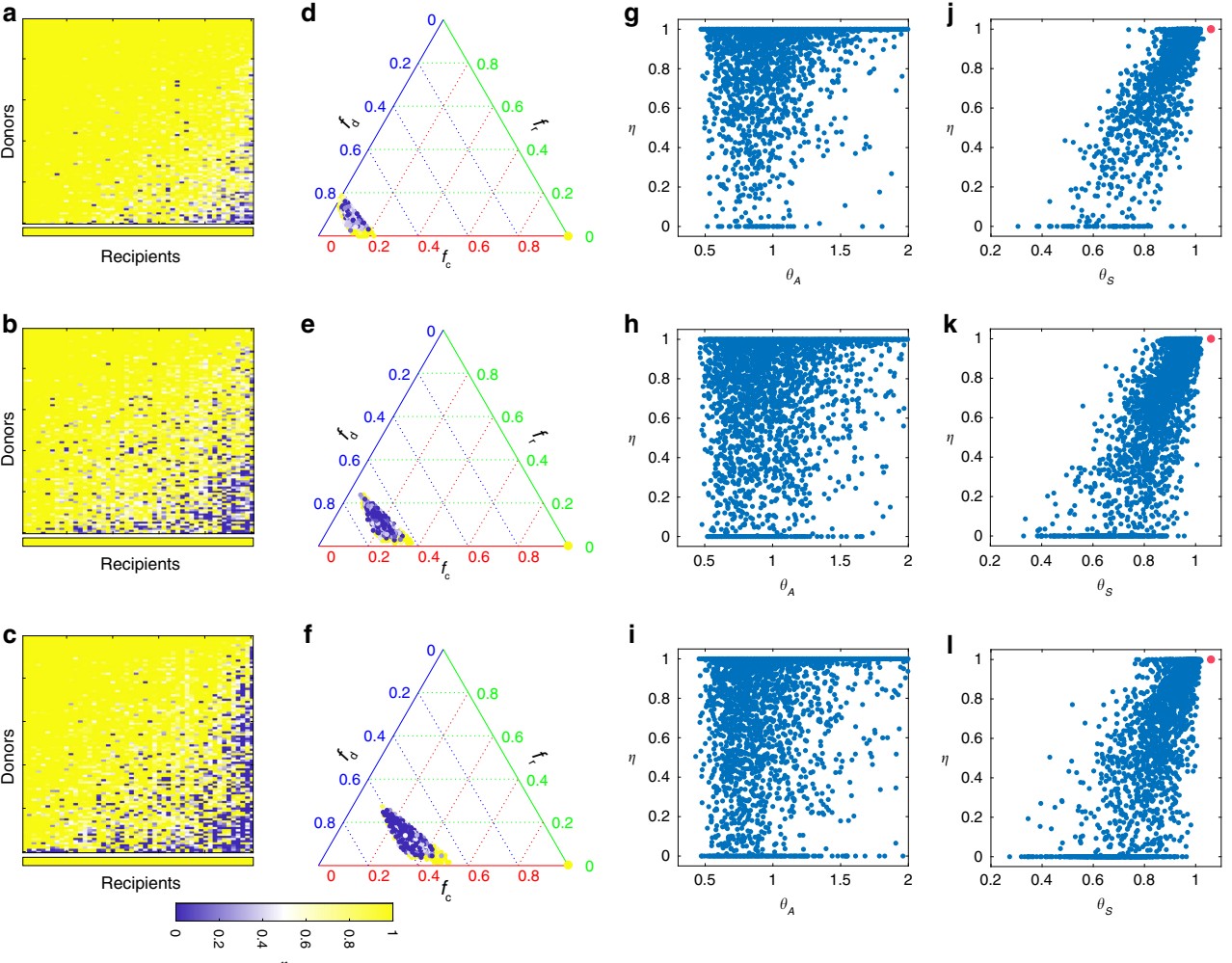

**Fig. 5 Impact of donor−recipient compatibility on FMT efficacy.** We simulate the FMT processes with universal microbial dynamics. Species in the gut microbiota of 100 healthy donors are randomly chosen from a pool of 100 species with the only condition that *C. difficile* cannot colonize. The resulting healthy microbiota of those 100 donors contains 60−80 species. From top to bottom, the pre-FMT microbiota of 50 recipients contain 10−15 (top), 20−25 (middle), and 30−35 (bottom) species, respectively. **a–c** We simulate the FMT process for each of the 100 × 50 (donor, recipient) pairs, and plot the recovery degree in a 100 × 50 matrix, where rows (or columns) are sorted based on the average recovery degree of each row (or column). The bottom row represents the recovery degree of autologous FMT, i.e., transplanting the recipient's own initially healthy microbiota back to his/her disrupted microbiota. **d–f** Recovery degrees of the 100 × 50 (donor, recipient) pairs in the ternary plot of fractions of donor-specific taxa ($f_d$), recipient-specific taxa ($f_r$), and common taxa ($f_c$). The color of each point represents the recovery degree of the FMT associated with the corresponding (donor, recipient) pair. The bottom right yellow point indicates the recovery degree of autologous FMT (with $f_c = 1$, $f_d = f_r = 0$). **g–i** Scatter plot of the recovery degree $\eta$ and the direct interaction strength ratio $\theta_A = \sum_{j=1,j\neq C}^{N} |a_{Cj}^-| / \sum_{j=1,j\neq C}^{N} a_{Cj}^+$. Here, $a_{Cj}^-$ for $a_{Cj} < 0$ ($a_{Cj}^+$ for $a_{Cj} > 0$) means direct inhibition (promotion) effect of species-$j$ on *C. difficile*. **j–l** Scatter plot of the recovery degree $\eta$ and the net or effective interaction strength ratio $\theta_S = \sum_{j=1,j\neq C}^{N} |s_{Cj}^-| / \sum_{j=1,j\neq C}^{N} s_{Cj}^+$. Here, $s_{Cj}^-$ for $s_{Cj} < 0$ ($s_{Cj}^+$ for $s_{Cj} > 0$) means net or effective inhibition (promotion) effect of species-$j$ on *C. difficile*. Red point indicates that in the post-FMT microbiota *C. difficile* becomes extinct, for which $\theta_S$ is undefined but the recovery degree $\eta = 1$. See Supplementary Note 1 for details on the model parameters used in the FMT simulations.

The intricate donor−recipient compatibility issue can again be partially explained by the network effect. To demonstrate this point, for each FMT simulation, we calculate $\theta_A = \sum_{j=1,j\neq C}^{N} |a_{Cj}^-| / \sum_{j=1,j\neq C}^{N} a_{Cj}^+$, representing the ratio between direct inhibition strengths ($a_{Cj}^-$ means $a_{Cj} < 0$) and direct promotion strengths ($a_{Cj}^+$ means $a_{Cj} > 0$) of other species on *C. difficile* in the recipient's post-FMT microbiota. We find no strong correlation between the direct strength ratio $\theta_A$ and the recovery degree $\eta$ (Fig. 5g–i). Then, for each FMT simulation, we calculate $\theta_S = \sum_{j=1,j\neq C}^{N} |s_{Cj}^-| / \sum_{j=1,j\neq C}^{N} s_{Cj}^+$, representing the ratio between net inhibition strengths ($s_{Cj}^-$ means $s_{Cj} < 0$) and net

promotion strengths ($s_{Cj}^+$ means $s_{Cj} > 0$) of other species on *C. difficile* in the recipient's post-FMT microbiota. (See Supplementary Note 2 and Supplementary Figs. 6 and 7 for the detailed calculation and graphical interpretation of the net or effective interactions $s_{ij}$ from the direct interaction $a_{ij}$ in the GLV model. Note that hereafter we call $S = (s_{ij}) \in \mathbb{R}^{N \times N}$ the contribution matrix, because its element $s_{ij}$ denotes the net contribution of species-$j$ on the steady-state abundance of species-$i$ in the community.) Now we see a strong correlation between the net strength ratio $\theta_s$ and the recovery degree $\eta$ (Fig. 5j–l). Simulation results shown here clearly demonstrate that net (rather than direct) impacts of the transplanted microbiota on *C. difficile*

strongly affects the FMT efficacy. This emphasizes the importance of understanding the ecological network of the human gut microbiota to design better microbiota-based therapeutics.

Since the ecological network of the human gut microbiota has not been mapped yet, finding the most compatible donor for a given recipient is very challenging (if not impossible), especially when the recipient has very high taxonomic diversity in the pre-FMT microbiota. A simple way to resolve this donor−recipient compatibility issue would be to perform autologous FMT[80]. In other words, a healthy individual would store her/his own fecal samples in a stool bank for future FMT use. This is conceptually similar to cord blood banking that stores umbilical cord blood for future use. In our simulations, we find that autologous FMT will always yield high recovery degree, regardless of the taxonomic diversity in the pre-FMT microbiota (see bottom rows in Fig. 5a–c).

**Design probiotic cocktails to decolonize pathogens**. The potential long-term safety concerns[81] and the challenging donor recruitment and screening process[22] have significantly limited the use of FMT. The development of probiotic cocktails containing only the effective components of FMT would alleviate these drawbacks largely due to the undefined nature of fecal preparations. However, such formulations attempted to-date have either not yet been tested or have failed clinical trials[82].

There are several challenges down the road. First, targeting microbes that directly inhibit the pathogen might backfire. For example, as shown in Fig. 6a, species-2 has a direct negative impact on the growth of *C. difficile*; however, if we just introduce species-2 to the diseased state, the abundance of *C. difficile* will become even higher than that of the diseased state. This is because, in the presence of mediator species-3, the net impact of species-2 on *C. difficile* is actually positive. This is a typical network effect. Since microbes rarely live in isolation but tend to aggregate into complex ecosystems[83–86], naive microbiota perturbations can ripple through the underlying ecological network resulting in unexpected and unwanted outcomes. The complex ecological network of the human gut microbiota must be accounted for to rationally design probiotic cocktails. But we haven't successfully mapped this ecological network yet.

Second, a magic bullet (i.e., a unique combination of microbial species) that works for all patients very likely does not exist. Indeed, different patients might have quite different species present in their diseased microbiota, and because the net impact of a species on *C. difficile* is context-dependent, there will be no generic probiotic cocktail that works for all patients. For example, as shown in Fig. 6a, b, the same probiotic cocktail {2, 10} could have quite different performance in decolonizing *C. difficile* for different patients. To design a truly personalized probiotic cocktail that works for a specific patient, we have to take into account the patient's diseased microbiota. To our knowledge, this has not been considered in any clinical trials.

Once we know the ecological network of the human gut microbiota, as well as the diseased microbiota of a patient, we can formalize an optimization problem to design a truly personalized probiotic cocktail. Our key idea is to calculate the net impact of a tentative probiotic cocktail on the growth of *C. difficile* and keep refining it by removing those species that could have a positive net impact on the growth of *C. difficile* in the altered microbial community. The iterative nature of our algorithm might sound like a trial-and-error approach. Here, we emphasize that this is really not the case, because we systematically considered the network effect during the iterations (see Supplementary Note 3.1 and Supplementary Fig. 8 for the details of our algorithm). For example, to decolonize *C. difficile* in the diseased state shown in

Fig. 6a, we first form a tentative probiotic cocktail containing all the effective inhibitors calculated from the global ecological network. Note that effective inhibitors include both direct and indirect inhibitors. But any species that already exists in the patient's diseased microbiota will be removed from the cocktail. The initial cocktail includes two direct inhibitors (species 2 and 10) and five indirect inhibitors (species 5, 9, 11, 12 and 13). Then for each species in the cocktail, we numerically test if it is still an effective inhibitor (i.e., has a negative net impact on the growth of *C. difficile*) in the altered local community (that contains all species in the patient's diseased microbiota and all species in the current cocktail). If yes, we keep it in the cocktail; if no, we remove it. We repeat this process until all the species in the cocktail are indeed effective inhibitors in the altered local community. Finally, we are left with a minimal set of species, i.e., the optimal probiotic cocktail $R_{global} = \{5, 9, 10, 11, 12, 13\}$, which can effectively inhibit the growth of *C. difficile* for this particular patient. Here the subscript "global" indicates that the cocktail was designed based on the global ecological network. As shown in Fig. 6a, the performance of this personalized probiotic cocktail is even better than that of the autologous FMT. Applying the same algorithm to another patient (with diseased state shown in Fig. 6b), we obtain another optimal probiotic cocktail $R_{global} = \{2, 5, 9, 10, 11\}$. Note that the two optimal probiotic cocktails are patient-specific, because they are designed based on the present species in each patient's diseased microbiota. In Fig. 6c, we show the designed optimal probiotic cocktails for 50 patients with a pool of 100 species for the metacommunity. Local communities corresponding to the patients' diseased gut microbiota are assembled as described in the previous section. Clearly, we see that optimal probiotic cocktails are patient-specific. Moreover, all of them can successfully suppress the growth of *C. difficile* with a recovery degree $\eta \approx 1$.

Even if we don't know the global ecological network of the human gut microbiota, knowing the ego network of *C. difficile* can still help us design a near-optimal personalized probiotic cocktail to decolonize *C. difficile* from the diseased microbiota of a particular patient. Here the ego network of *C. difficile* consists of a focal node/species (ego, i.e., *C. difficile*), those nodes/species to which *C. difficile* directly interact with (they are called alters), the links/interactions between *C. difficile* and its alters, as well as the links/interactions among the alters. The algorithm to design a probiotic cocktail based on the ego network of *C. difficile* is very similar to the algorithm based on the global ecological network. The only difference is that we need to construct the initial tentative probiotic cocktail based on the ego network. In other words, we need to consider all the effective inhibitors calculated from the ego network, instead of the global ecological network (see Supplementary Note 3.2 for details). For the diseased state shown in Fig. 6a, we find that the ego network-based cocktail, $R_{ego} = \{10\}$, can indeed suppress the abundance of *C. difficile* to a much lower level than that of the diseased state. For the diseased state shown in Fig. 6b, the ego network-based cocktail $R_{ego} = \{2, 10\}$ actually works almost equally well as the optimal cocktail $R_{global} = \{2, 5, 9, 10, 11\}$ designed based on the global ecological network. Note that the algorithm can be generalized to consider the $k$-step ego network instead of the 1-step ego network used in Fig. 6a, b (see Supplementary Fig. 9). For the same set of 50 patients as shown in Fig. 6c, we calculate the ego-network-based personalized probiotic cocktails (Fig. 6d), finding that most of them can successfully suppress the growth of *C. difficile* with recovery degree $\eta \approx 1$. This clearly demonstrates the near-optimality of the ego-network-based probiotic cocktails. Since inferring the ego network of *C. difficile* should be much easier than inferring the global ecological network of the human gut

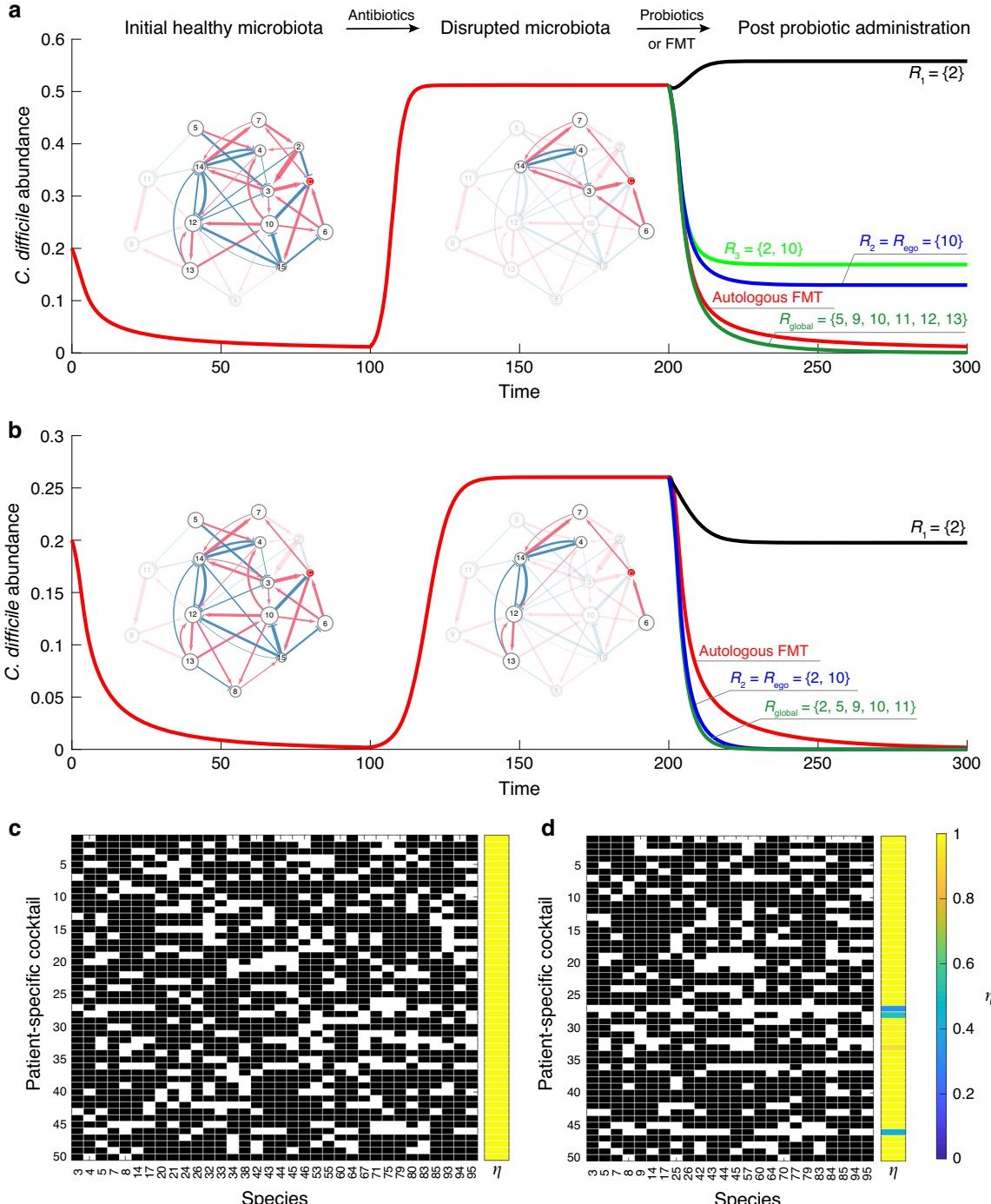

**Fig. 6 Personalized probiotic cocktails for the decolonization of *C. difficile*. a, b** Different combinations of species, i.e., probiotic cocktails, can have drastically different performance in suppressing the overgrowth of *C. difficile*. We start from an initial healthy microbiota (represented by a subgraph of a global ecological network with 15 species). Antibiotic administration removes certain species (especially those can directly inhibit the growth of *C. difficile*), leading to a disrupted microbiota (diseased state). Note that different patients could have totally different disrupted microbiota, rendering the design of a generic probiotic cocktail infeasible. **c** The personalized optimal probiotic cocktails of 50 patients designed from a global ecological network of 100 species. The model parameters used in the simulations of panels (**c**) and (**d**) are described in Supplementary Note 1. **d** The personalized near-optimal probiotic cocktails of the same set of 50 patients designed from the ego network of *C. difficile*.

microbiota, the simulation results presented here hold promise for alleviating the difficulty in the rational design of personalized probiotic cocktails in treating rCDI.

**Network effect in a mouse microbial community.** In our modeling framework, we proposed the concept of network effect

and emphasize that the net impact of a species on the growth of another species is largely context-dependent. Specifically, if we compare the direct impact with the net impact, there are three cases: (i) normal: the direct and net impacts share the same sign; (ii) bridging: the direct impact is zero while the net impact is not; and (iii) counter-intuitive: the direct and net impacts have opposite signs.

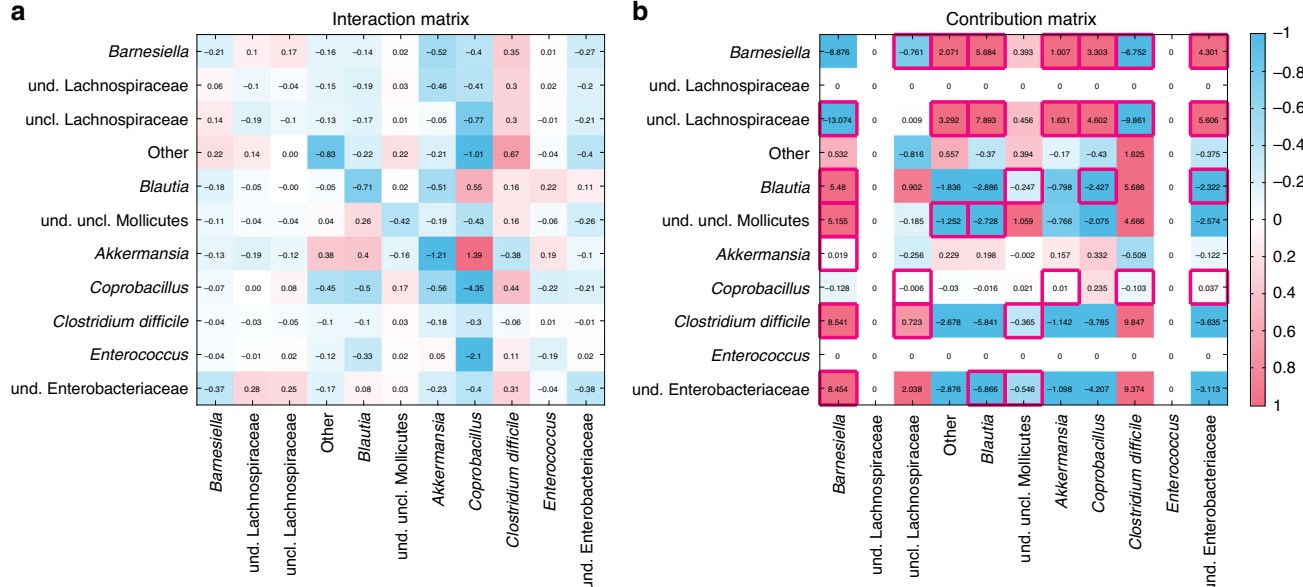

**Fig. 7 Interaction matrix and contribution matrix of a real microbial community. a** The interaction matrix of an ecological network inferred from mice experiments on antibiotic-mediated CDI[66]. **b** The contribution matrix of this ecological network. Zero rows and columns indicate two taxa that will go to extinction asymptotically in this community (see Supplementary Fig. 3 for details). The red boxes indicate the counter-intuitive cases of the network effect. For the visualization purpose, the color scale was set from −1 to 1, and the values <−1 (or >1) were indicated as the same color of the value −1 (or 1), respectively.

To directly demonstrate the presence of network effect, we analyzed the microbial interaction matrix (Fig. 7a) inferred from the mouse experiments of antibiotic-mediated CDI[66]. The experiments consisted of three populations of mice: (i) The first population received spores of *C. difficile*, and was used to determine the susceptibility of the native microbiota to invasion by the pathogen. (ii) The second population received a single dose of clindamycin to assess the effect of the antibiotic alone. (iii) The third population received a single dose of clindamycin and, on the following day, was inoculated with *C. difficile* spores. A GLV model with an additional external perturbation (i.e., antibiotic) was used to model the ecological dynamics (e.g., intrinsic growth rates, inter-taxa interactions, etc.). From the inferred interaction matrix (Fig. 7a), we calculated the contribution matrix (Fig. 7b), where zero rows and columns represent taxa that cannot coexist with other taxa in equilibrium. Red boxes in Fig. 7b highlight the counter-intuitive cases of the network effect. This result underscores the importance of our ecological modeling framework to understand the efficacy of FMT.

**Pre-FMT taxonomic diversity in a clinical trial**. Our modeling framework predicted that the FMT efficacy is negatively correlated with the taxonomic diversity of the recipient's pre-FMT microbiota. To test our prediction, we analyzed real FMT data from a clinical trial[87], where in total 106 rCDI patients were treated with encapsulated donor material for FMT (cap-FMT). Figure 8a shows the sequenced fecal samples from 7 healthy donors and 88 rCDI patients at different time points: pre-FMT, 2–6 days post FMT, weeks (7–20 days) post FMT, months (21–60 days) post FMT, and long term (>60 days). (Fecal materials from some patients were not available for sequencing in this clinical trial.) Figure 8b shows the PCoA plot of those samples, from which it is hard to distinguish responders from nonresponders.

Interestingly, we found that nonresponders tend to have higher median taxonomic diversity than responders (Fig. 8c–e). This clinical evidence partially supports our simulation result that

FMT efficacy generally decreases with increasing taxonomic diversity of the pre-FMT microbiota. Note that the difference is not statistically significant. We anticipate that this might be due to the imbalance between sample sizes of responders ($n = 71$) and nonresponders ($n = 17$). Further clinical studies are definitely needed to validate our theoretical prediction.

**Donor–recipient compatibility in a clinical trial**. To demonstrate the donor–recipient compatibility issue using real data, we analyzed the microbiome samples from the cap-FMT clinical trial mentioned above[87]. Note that in this trial, one donor's fecal material was transplanted into many different recipients. As shown in Fig. 8a, for almost each of the donors, most of recipients responded to the cap-FMT, but a few recipients did not. For each (donor, recipient) pair, we further calculated the fractions of donor-specific taxa ($f_d$), recipient-specific taxa ($f_r$), and common taxa ($f_c = 1 - f_d - f_r$) at different time points. We found that it is impossible to distinguish responders and nonresponders in the ternary plot (Fig. 8f–j). This result is consistent with our simulation result shown in Fig. 5d–f.

**Design probiotic cocktails for a mouse microbial community**. In ref. [65], the ecological network involving the so-called Gnoto-Complex microflora (a mixture of human commensal bacterial type strains) and *C. difficile* was inferred from mouse data based on the assumption that the microbial community follows the GLV model. In particular, germ-free mice were first precolonized with the GnotoComplex microflora and the commensal microbiota were allowed to establish for 28 days. Then, mice were infected with *C. difficile* spores and monitored for an additional 28 days. From the ecological network (Fig. 9a), we notice that species-4 (*Clostridium scindens*) and species-13 (*Roseburia hominis*) can directly inhibit the growth of *C. difficile*, while species-1 (*Clostridium hiranonis*), species-3 (*Proteus mirabilis*), species-5 (*Ruminococcus obeum*), species-7 (*Bacteroides ovatus*), and species-12 (*Klebsiella oxytoca*) can indirectly inhibit the growth of *C. difficile* through some mediating species. Based on

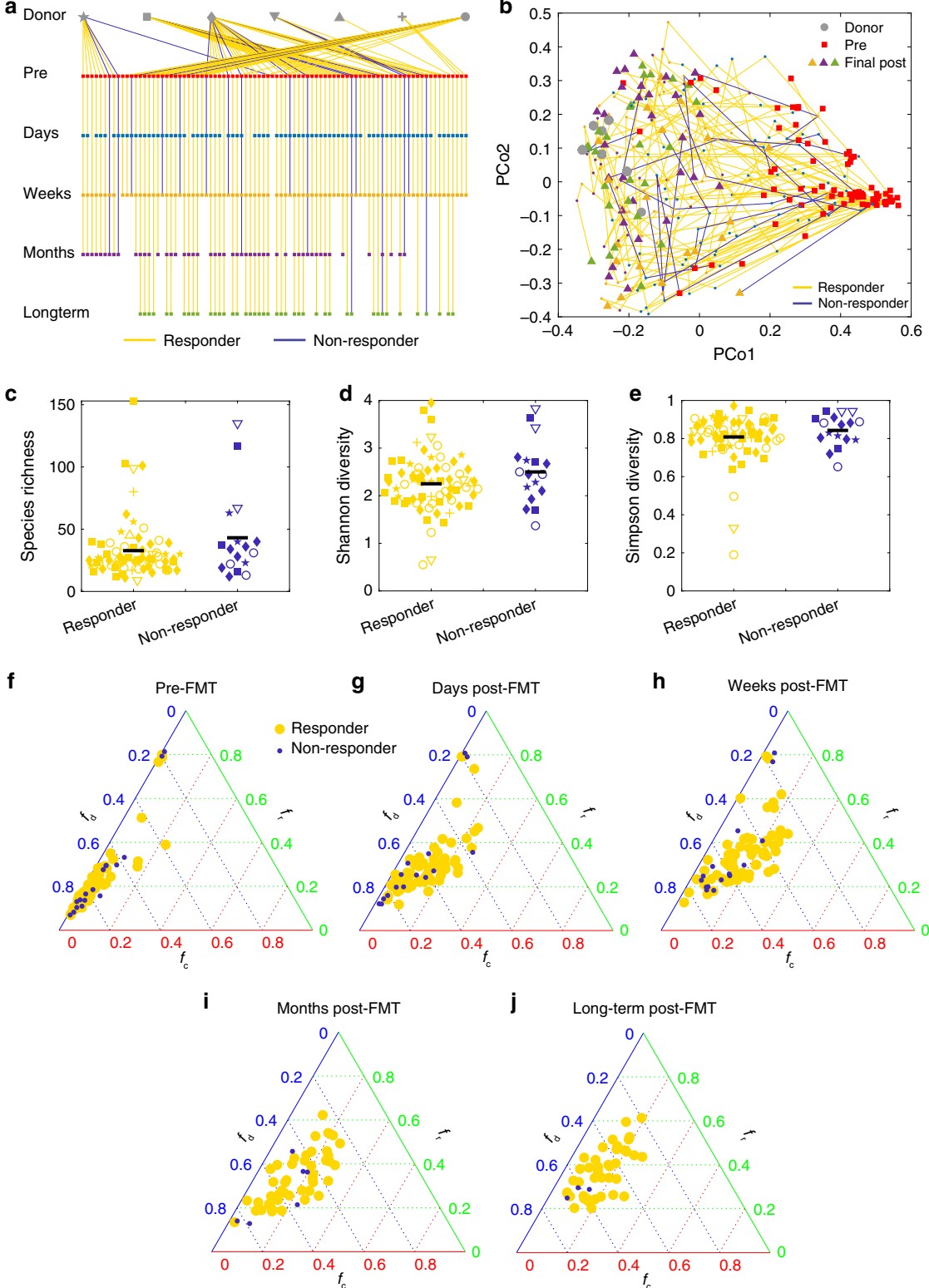

this ecological network and the disrupted microbiota, we applied our framework to design probiotic cocktails to effectively decolonize *C. difficile*.

In Fig. 9a–c, we showed the initial community composed of all the 14 species in the network, a disrupted microbiota due to hypothetic antibiotic administration, and the restored community after the administration of a particular probiotic cocktail, respectively. Figure 9b demonstrates the efficacy of various

probiotic cocktails. The optimal probiotic cocktail $R_{global}$ is designed based on the global ecological network (Fig. 9a) and the specific disrupted microbiota (Fig. 9b). This cocktail $R_{global}$ contains two direct inhibitors of *C. difficile* (i.e., species-4: *C. scindens* and species-13: *R. hominis*), and two indirect inhibitors of *C. difficile* (i.e., species-5: *R. obeum* and species-12: *K. oxytoca*). As shown in Fig. 9d (green curve), this cocktail $R_{global}$ can strongly suppress the abundance of *C. difficile*. Note that species-12

**Fig. 8 Pre-FMT taxonomic diversity and donor−recipient compatibility in a clinical trial. a** In this clinical trial[87], the fecal material of each donor was used in FMT for different recipients. For a typical donor, some recipients responded to FMT (yellow lines), some did not (blue lines). The trial collected samples at different time points: pre-FMT, 2–6 days post FMT, weeks (7–20 days) post FMT, months (21–60 days) post FMT, and long term (>60 days). **b** The trajectories of recipients' samples from pre-FMT to final post-FMT are visualized in the PCoA plot (using Bray–Curtis dissimilarity between samples at genus level). Pre-FMT, post-FMT and final samples are represented as squares, small dots, and triangles, respectively, with colors indicating the different time points as shown in in panel (**a**). The taxonomic diversity of the responders' and nonresponders' pre-FMT microbiota at the OTU level are compared by using three indices: **c** species richness ($p = 0.18$), **d** Shannon diversity ($p = 0.13$), and **e** Simpson diversity ($p = 0.28$). There are 71 responders and 17 nonresponders. Hypothesis testing for differences of the means were done by a linear mixed effects analysis using treatment as fixed effects and donor ID as a random effect. The linear mixed model was fit to data via REML (restricted maximum likelihood), using the lme4 package in R. The $p$ values were computed via the Satterthwaite's method, using the lmerTest package in R. The black line represents mean value of the points. The shape of each data point in **c**–**e** is consistent with that of the recipient's corresponding donor as shown in panel (**a**). **f**–**j** Ternary plot of fractions of donor-specific taxa ($f_d$), recipient-specific taxa ($f_r$), and common taxa ($f_c$) for each (donor, recipient) pair at different time points. **f** Pre FMT. **g** Days post FMT. **h** Weeks post FMT. **i** Months post FMT. **j** Long-term post FMT.

(*K. oxytoca*) is actually an opportunistic pathogen[88]. Due to safety concerns, we should exclude it from any cocktail. It turns out the near-optimal cocktail $R_{\text{near-optimal}}$, obtained by excluding *K. oxytoca* from $R_{\text{global}}$, can still decolonize *C. difficile* to a large extent (see red curve in Fig. 9d). We also designed several cocktails based on the $n$-step ego-networks of *C. difficile* (with $n = 1, 2, 3$), which just contain those species that are $n$-step away from *C. difficile* in the original network (Fig. 9a). Note that for this small network, $R_{\text{ego-1}}$ is the same as the cocktail $R_d$ designed by only considering the direct inhibitors, while $R_{\text{ego-3}}$ is actually identical to $R_{\text{global}}$. Moreover, $R_{\text{ego-1}}$ and $R_{\text{ego-2}}$ just represent two subsets of $R_{\text{global}}$. Though the performance of $R_{\text{ego-1}}$ and $R_{\text{ego-2}}$ are not comparable with $R_{\text{ego-3}} = R_{\text{global}}$, they both can suppress the abundance of *C. difficile* to a much lower level than that of the diseased state. For comparison purposes, we also showed the performance of three other cocktails ($R_1$, $R_2$ and $R_3$), representing three randomly chosen subsets of $R_{\text{global}}$. We found that none of them is comparable with $R_{\text{global}}$. We emphasize that $R_{\text{global}}$ is designed based on the specific disrupted microbiota, hence it is personalized. For a different disrupted microbiota (e.g., as shown in Fig. 9f), we can design a different $R_{\text{global}}$, which again outperforms any other cocktails (Fig. 9h). Overall, these results demonstrate the advantages of our network-based design of personalized probiotic cocktails.

## Discussion

The status quo as it pertains to the ecological understanding of FMT as an effective microbiome-based therapy in treating rCDI can be summarized as: little or no understanding of the treatment at the systems level. That has been the case despite numerous studies that have been conducted to reveal the efficacy of FMT. For example, several observational studies described changes in the associated microbial community structure[19,20,89]. Some mechanistic studies have shown that FMT restores secondary bile acid metabolism[50], valerate[51] and microbial bile salt hydrolases mediate the efficacy of FMT[52], and the restored microbiota can inhibit *C. difficile* through competition for nutrients, antimicrobial peptides, and activation of immune-mediated colonization resistance[53]. A machine-learning method to predict the gut microbiota of the post-FMT patients was also developed, using both clinical and metagenomics data of the donor and the pre-FMT patient[79]. Several multicenter studies[58,90] focused on the identification of clinical risk factors that are associated with FMT failure (without using microbiome data).

Our approach represents a new and substantive departure from the status quo by shifting the focus from specific species or functions to a systems-level understanding of the human gut microbiome using community ecology theory (rather than relying on any machine-learning techniques). In particular, we use an ecological modeling framework to predict a few key factors that determine the success of FMT in treating rCDI. We also propose a network-based method for the rational design of truly personalized probiotic cocktails for the treatment and preventing of rCDI. Though we choose rCDI as a prototype disease, the presented results could have implications in other diseases associated with a disrupted microbiome and will have a positive translational impact on the development of general microbiota-based therapeutics.

We emphasize that there are several limitations in our current modeling framework. First, stochastic effects are considered negligible. In principle, we can incorporate stochastic effects in our model and use the resulting stochastic differential equations to simulate the FMT process. Based on our previous numerical studies on the origins and control of community types in human microbiome[91], we anticipate that this will not change our main results on the key factors determining the success of FMT. Second, our current modeling framework does not explicitly model the dynamics of resources provided to and/or chemicals secreted by the microbial species[92–97]. Hence, our framework does not offer insights designing prebiotics that can effectively restore healthy microbiota. We consider this will be a natural extension of our current modeling framework and deserves dedicated efforts in a future work. Third, the current modeling framework starts with a minimal dynamical model of taxonomic abundances to facilitate the parameterizing procedure and thus does not reflect functional changes during the FMT. Further efforts should be dedicated to integrate both taxonomic and functional data to provide a more comprehensive modeling framework. Moreover, such an integrative modeling/analysis of taxonomic and functional data will enable us to better design personalized probiotic cocktails. Indeed, though consisting of different combinations of species, those personalized probiotic cocktails could restore the microbial functions (such as secondary bile acid metabolism) in a generic way, thanks to the functional redundancy of microbial species. Fourth, the current modeling framework does not take into account the impact of dietary intake and drugs on the host's microbial composition. Finally, the quantification of net impact of a species on the growth of *C. difficile* and the design of optimal personalized probiotic cocktails are largely based on the GLV model (which assumes linear functional response and pair-wise microbial interactions). For more complicated population dynamics models with nonlinear functional response or higher-order interactions, it is still an open question how to analytically calculate the net impact.

Of course, carefully designed animal experiments and clinical trials will be needed to further validate our theoretical predictions. Artificial guts (such as the gut-on-a-chip[98] and the HuMiX[99] system) would also be intriguing to test our predictions, though an important challenge still lies in further increasing their high-throughput analyses capacity[100]. We hope this work will catalyze more collaborative works between modelers, microbiologists, and clinicians.

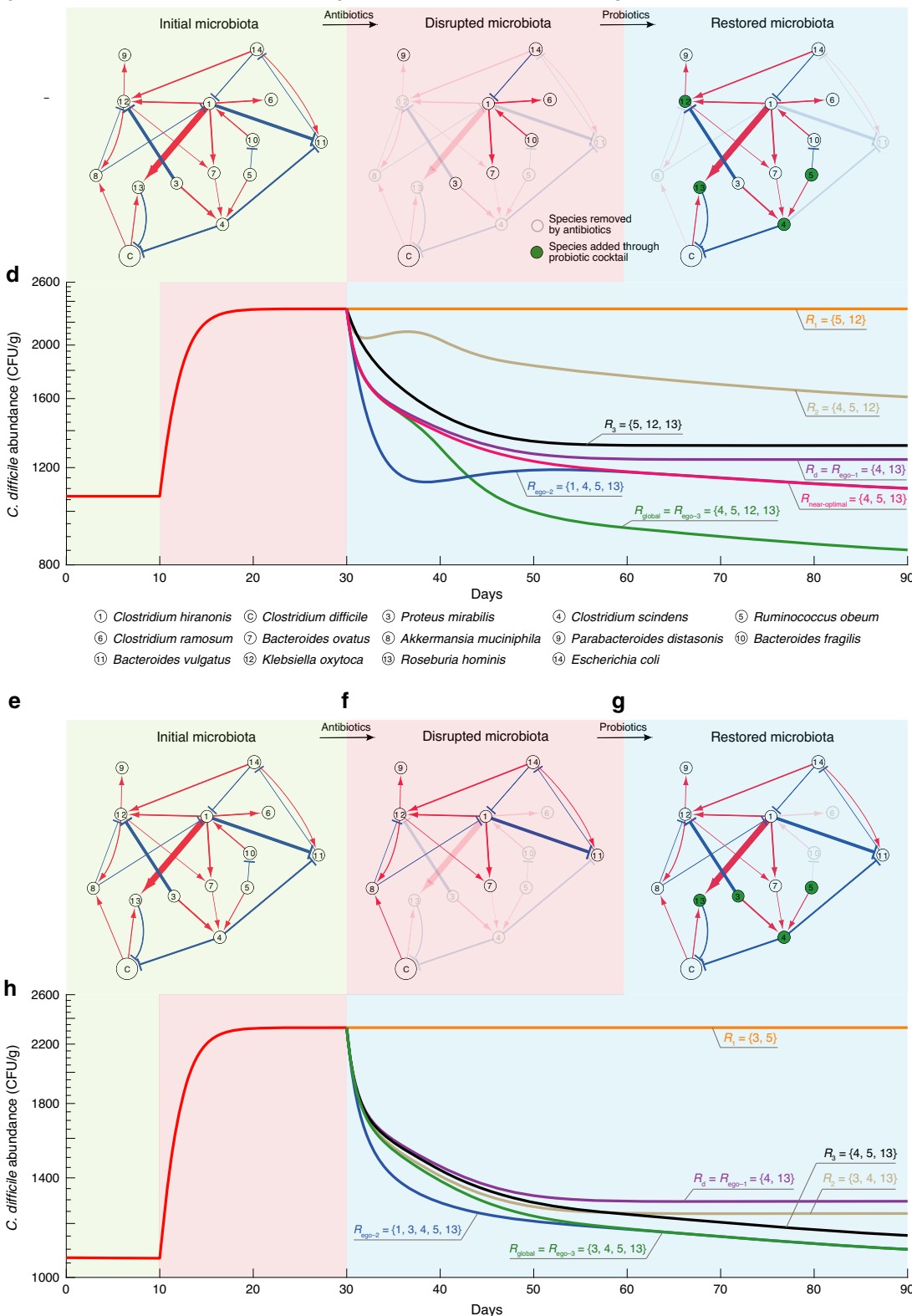

1. *Clostridium hiranonis*  C. *Clostridium difficile*  3. *Proteus mirabilis*  4. *Clostridium scindens*  5. *Ruminococcus obeum*
6. *Clostridium ramosum*  7. *Bacteroides ovatus*  8. *Akkermansia muciniphila*  9. *Parabacteroides distasonis*  10. *Bacteroides fragilis*
11. *Bacteroides vulgatus*  12. *Klebsiella oxytoca*  13. *Roseburia hominis*  14. *Escherichia coli*

## Methods

**Population dynamics of microbial communities**. A detailed dynamic model of the human microbiome would have to include mechanistic microbial interactions, spatial structure of the particular body site, as well as host–microbiome interactions. For the sake of simplicity, in our modeling framework we focus on exploring the impact that any given microbe has on the abundance of other microbes. To achieve that, we consider a phenomenological population dynamics model in the form of:

$$\mathrm{d}x_i(t)/\mathrm{d}t = x_i(t)\left[r_i + \sum_{j=1}^{N} a_{ij} g\left(x_i(t), x_j(t)\right)\right], i = 1, \ldots, N. \quad (1)$$

**Fig. 9 Personalized probiotic cocktails effectively decolonize *C. difficile*. a** An ecological network involving the GnotoComplex microflora (a mixture of human commensal bacterial type strains) and *C. difficile* was inferred from mouse data[65]. Node C represents *C. difficile*. The edge width and node size indicate the interspecies interaction strength and the intrinsic growth rate, respectively. Red (or blue) edges indicate the direct promotion (or inhibition), respectively. **b** A disrupted microbiota due to a hypothetic antibiotic administration. **c** The restored microbiota due to the administration of a particular probiotic cocktail $R_{global}$. **d** The trajectory of *C. difficile* abundance over three different time windows: (1) the initial healthy microbiota, (2) the disrupted microbiota, and (3) the microbiota post probiotic administration. In the third time window, we compare the performance of various probiotic cocktails in terms of their ability to decolonize *C. difficile*. Those cocktails were designed by considering direct inhibitors only ($R_d$), the global ecological network ($R_{global}$), the *n*-step ego-networks of *C. difficile* with *n* = 1, 2, 3 ($R_{ego-1}$, $R_{ego-2}$, and $R_{ego-3}$), and randomly chosen subsets of $R_{global}$ ($R_1$, $R_2$, and $R_3$). $R_{near-optimal}$ is obtained by excluding species-12 (i.e., *K. oxytoca*, which is an opportunistic pathogen) from $R_{global}$. Note that $R_{ego-1}$ is not necessarily always the same as $R_d$ because $R_{ego-1}$ needs to consider the net impact among neighbors of the focal species. **e–g** We start from the same initial microbiota as shown in panel (**a**). But another hypothetic antibiotic administration can lead to a different disrupted microbiota (**f**), which can be restored through probiotic administration (**g**). **h** Performance of different probiotic cocktails in decolonizing *C. difficile* vary.

Here, $x_i(t)$ denotes the abundance of species-*i* at time *t*, $r_i \in \mathbb{R}$ is the intrinsic growth rate of species-*i*, $a_{ij}$ (when $i \neq j$) reflects the type of direct impact that species-*j* has on the population change of species-*i*, i.e., $a_{ij} > 0$ (<0, or =0) means that species-*j* promotes (inhibits, or does not affect) the growth of species-*i*, respectively. We define a matrix $A = (a_{ij}) \in \mathbb{R}^{N \times N}$ to present all the pair-wise microbial interactions. The ecological network $\mathcal{G}(A)$ is the graph representation of the interaction matrix *A*: there is a directed edge $(j \rightarrow i) \in \mathcal{G}(A)$ if and only if $a_{ij} \neq 0$. Various methods have been developed to infer the microbial interactions and map the ecological network $\mathcal{G}(A)$ from time series[65,91] or steady-state data[101]. In Eq. (1) the function $g(x_i, x_j) : \mathbb{R} \times \mathbb{R} \rightarrow \mathbb{R}$ is the so-called functional response in community ecology, which models the intake rate of a consumer as a function of food density. There are different functional responses with different levels of complexity, representing different mechanisms of interspecies interactions. The simplest case is a linear functional response $g(x_i, x_j) = x_j$ for which Eq. (1) reduces to the classical GLV model[66,83]. In this model, $a_{ij}$ (when $i \neq j$) accounts for the direct impact (i.e., interaction strength) that microbe *j* has on the population change of microbe *i*, and the terms $a_{ii}x_i^2$ are adopted from Verhulst's logistic growth model[102].

**Reporting summary**. Further information on research design is available in the Nature Research Reporting Summary linked to this article.

## Data availability

Data analyzed in this work are available at https://github.com/xiaoyandong08/FMT_simulation_framework. Source data are provided with this paper.

## Code availability

Matlab-R2018a and R-3.5.2 code used in this work are available at https://github.com/xiaoyandong08/FMT_simulation_framework.

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

## Acknowledgements

We thank Professor Michael J. Sadowsky for kindly sharing microbiome data of the cap-FMT clinical trial. Y.-Y.L. acknowledges grants from National Institutes of Health (R01AI141529, R01HD093761, UH3OD023268, U19AI095219, and U01HL089856). Y.X. acknowledges funding support from China Scholarship Council (CSC) and National Natural Science Foundation of China (Grant No. 61902418). M.T.A. gratefully acknowledges the financial support from CONACyT project A1-S-13909, México.

## Author contributions

Y.-Y.L. conceived and designed the project. Y.X. performed all the analytical/numerical calculations and real data analysis under the supervision of Y.-Y.L. All authors analyzed the results. Y.-Y.L. and Y.X. wrote the manuscript. M.T.A., S.L. and S.T.W. edited the manuscript.

## Competing interests

The authors declare no competing interests.
