## [Peer Review File · Nature Communications]

Reviewers' comments:

Reviewer #1 (Remarks to the Author):

To import concepts and approaches from ecology is an excellent idea, not totally new, but still mostly unexplored. Designing optimal cocktails is a very interesting idea, and to do this based on systems analysis is really promising. Theoretically speaking, the paper is very wise and perspectival. The question is feasibility and applicability, see my comments below, both minor and major.

Figure 1, legend: "high taxonomic diversity", yes, this is true but the key point, according to the ecological approach of the paper, is high functional diversity.

"dynamics analysis" could be changed to "dynamical analysis"

In passing, it would be a lot easier for the reviewer to use line numbering....

"When modeling a dynamical ecosystem, we first decide how complex the model needs to be so as to capture the phenomenon of interest." - this sentence is more like in textbook style, I suggest to remove this.

"Microbial interactions can be mediated by direct secretion of substances such as bacteriocins^{53,54}, competition for the same nutrient⁵⁵, or immune system modulation^{56,57,58}." - interactions between bacteria X and Y are really of composite nature, so the nature of the ultimate relationship is a black box. Also, to describe all these interactions experimentally is not feasible. This is why statistical associations are typically used for inferring positive and negative interactions. These can be used to parameterize dynamical models but the question is how reliable is this approach. You may elaborate these limitations.

Figure 2: the framework is nice and the inspiration from ecological literature is great. But only theoretically. Because of the above, no one can parameterize these interactions today.

Moreover, theoretical results should be tested experimentally. Artificial guts exist but their readiness for community-level tests should be briefly mentioned or discussed.

Figure 2 legend: change "Principle" to "Principal"

I am not sure how the steady state condition can be understood. The mammal gut microbiome is a highly dynamical system, sensitive to environmental effects and actual food consumption. To what extent can we speak about steady state or any kind of balance? What is the time-scale for this?

It is unclear what is meant by strong universality of gut microbiome samples. There is huge variability exactly in the HMP database. Some systemic outputs can be more general.

"To find a compatible donor for a given recipient, one may consider a naive approach: compare their taxonomic profiles, and calculate the fractions of donor-specific taxa (denoted as f_d), recipient-specific taxa (f_r), and common taxa ($f_c=1-f_d-f_r$)." - this approach is a good one but it dramatically reduces the holistic, systemic information provided by the network approach.

The difference between cord blood banking and the idea of fecal sample banking is that animal development is a much more conservative process, while the development (assembly) of the microbiota is very much exposed to environmental effects. I would trust more in variability and selection (trial and error) between different treatments.

To potentially improve the link to ecological literature, let me suggest some further references:

FMT sounds more like an invasion meltdown situation, see
<https://esajournals.onlinelibrary.wiley.com/doi/abs/10.1890/11-0050.1>

For the relationship between invasion and community health, see the famous example of Mnemiopsis invasion of the Black Sea (disturbed community was more invadable):
<https://onlinelibrary.wiley.com/doi/abs/10.1046/j.1365-2419.1998.00080.x>

For identifying critically important elements in complex dynamical ecosystems, see:
<https://www.nature.com/articles/srep15920>

The suggested modelling approach remains a theoretical exercise for many species. A possibly more realistic approach would be to parameterize dynamical models and simulate them for small network modules of 3-4-5 species. In this case, interactions could even be measured experimentally and then realistic models could be built, following the same kind of ecological logic.

Reviewer #2 (Remarks to the Author):

The manuscript by Xiao et al. describes an ecological framework to study the efficacy of FMT to treat rCDI. This framework utilises changes in bacterial taxa pre- and post-FMT (and FMT donor taxa) to propose bacteria that are important for inhibiting *C. difficile* growth. This ecological framework is interesting and potentially a very useful tool to design personalised synthetic microbial communities to treat diseases. However, I'm not sure how applicable this work will be in the absence of functional data (especially in the case of CDI). Moreover, some of the claims made in this paper for rCDI do not align with what we know of the mechanisms of FMT to treat rCDI. Finally, the manuscript as it exists in its current form relies on simulated data, and needs to be validated with in vivo and in vitro data.

Comments:

- Lines 67-68: The authors write "In a healthy gut microbiota, *C. difficile* is "out-competed" by hundreds of strains of bacteria that are normally present (Fig.1a)." However, this implies the mechanism of FMT is competition for nutrients (which isn't precisely correct based on our current understanding of the mechanism of FMT for rCDI). Reword to "In a healthy gut microbiota, *C. difficile* is unable to colonise the gut in the presence of the hundreds of strains of bacteria that are normally present (Fig. 1a)." Or something similar.
- Line 70 (and throughout the manuscript, including Fig 1): change "flourish" to "grow"
- Line 72 (and throughout the manuscript): "flora" is an outdated term. "Microbiota" is the preferred term.
- Line 73: Change "and recurs in" to "and CDI recurs in"
- Lines 93-95: The authors state "Yet, we still don't understand why FMT works so well for rCDI and why ~20% of rCDI patients relapse after FMT. In fact, very little is known about the underlying mechanism of FMT and its long-term effects from the ecological perspective." I disagree. There have been several studies that demonstrate the mechanism of FMT to treat rCDI. I would reword to say that we have only recently started to understand the mechanisms of FMT to treat rCDI.
- Line 94: Missing a key reference that describes the mechanism of FMT to treat CDI. Add the following reference: McDonald JAK et al. (2018) Inhibiting Growth of *Clostridioides difficile* by Restoring Valerate, Produced by the Intestinal Microbiota. *Gastroenterology*. 155(5):1495-1507.

- Lines 97-98: The authors state “Why does FMT work better for recurrent CDI than for primary CDI?” FMT is used for rCDI more than primary CDI because it is not a first line treatment for primary CDI (antibiotics are the first line treatment). Has anyone ever compared the efficacy of FMT to treat primary CDI vs rCDI? If so, a reference is needed here.
- I cannot comment on the mathematical equations described in this manuscript, as this is outside of my field (lines 117-243 and lines 421-432).
- Lines 233-235: When a gut microbiota is disrupted by antibiotics, it not only results in killing of members of a healthy gut microbiota, but also the bloom of members of the healthy gut microbiota that are resistant to the antibiotic. Does your ecological framework take this into account, or is it only presence/absence?
- Lines 257-259: The authors state “For example, in treating rCDI with FMT, an extreme scenario could be that the same set of microbes that effectively inhibit the growth of *C. difficile* in the donor’s gut might promote the growth of *C. difficile* in the recipient’s gut.” This is very unlikely. FMT banks that provide stool from the same donor to multiple rCDI patients have not demonstrated this. In fact, FMT has never been shown to promote CDI (FMT is very effective for treating rCDI, regardless of the FMT donor’s gut microbiota composition).
- Line 269: What do the authors mean by the term “universality”?
- Lines 271-275: The gut microbiota of rCDI patients are variable because the patient cohort of quite heterogeneous: many different kinds of patients (on a variety of diets/medications), all receiving a wide variety of antibiotics that result in a first episode of CDI, then have varying recovery following antibiotics before they develop CDI, then who go on to receive a broad spectrum antibiotic to treat CDI, then who have varying recovery responses before developing rCDI, who then get more broad spectrum antibiotics before receiving FMT. In addition to the varying starting composition of their gut microbiota, they have different histories. So it’s not unexpected that rCDI patients do not have similar community dynamics.
- Lines 343-345: The authors state “It is reasonable to assume that primary CDI patients have less dysbiotic and more diverse gut microbiota than that of rCDI patients.” This has been demonstrated in a published study (add this reference): Allegretti JR, et al. (2016) Recurrent *Clostridium difficile* infection associates with distinct bile acid and microbiome profiles. *Aliment Pharmacol Ther.* 43(11):1142-53.
- Lines 359-361: Another (more probable) interpretation is that when you add more species, either they have no effect on *C. difficile* growth, or they have functional redundancy (so adding them makes no difference). Or, if adding more species reduces FMT efficacy, then it could be that a species that does not affect *C. difficile* growth is inhibiting the growth of a species that can inhibit *C. difficile* growth.
- Lines 368-370: A more probable explanation is that the donor strains could not engraft because of competition between the recipient’s existing bacteria and the bacteria in the FMT. Based on what we know about the mechanism of FMT to treat CDI I do not believe bacteria in the FMT donor would promote *C. difficile* growth in a CDI patient.

- Lines 393-395: But rCDI patients don't have high diversity of their gut microbiota. How do you define high diversity in these patients? And can you confirm that this patient was colonised with vegetative *C. difficile* (not spores) at the time the sample was collected? Do you have data to compare the levels of *C. difficile* in patients with low diversity to levels found in patients with high diversity (not 16S rRNA gene sequencing data, but qPCR levels or plate counts)?
- Lines 477-480: I am not convinced by this argument based on what we know about the mechanism of FMT to treat rCDI. We cannot rely on 16S data alone, but we must also look at changes in the function of the community. This would greatly strengthen this model.
- Lines 487-490: But FMT works well to treat rCDI, regardless of the donor and how the FMT was prepared. This suggests restoration of a microbial function (that has redundancy across species) is responsible for treating CDI. This is supported by what we understand about the mechanism of FMT to treat rCDI.
- Lines 511-513: This is too simplistic, as some microbial processes require many species to work together to carry out a function. You cannot simply look at which species are directly inhibiting *C. difficile*, as you may need additional species to carry out some steps first.
- Much of this work relies on simulated data sets. These methods need validation on published data sets.

Reviewer #3 (Remarks to the Author):

I have now read "An Ecological Framework to Understand the Efficacy of Fecal Microbiota Transplantation" by Liu and colleagues. They present computer simulations of microbial ecosystem dynamics using several established ecological models. The authors use these equations to study how the rise of *C. difficile* (following a simulated dose of antibiotics) may be remedied with a simulated fecal microbiota transplantation.

The gut microbiota community is modeled as sets of differential equations representing the growth rates of individual species. *C. difficile* is one of these species. The authors investigate under what conditions the abundance of *C. diff* is reduced and a pre-antibiotic community is restored. They also consider the potential for the between-species interactions to differ between hosts. The authors then simulate the development of species abundances over time. Simulating ODEs can be educational, and the authors here show that depending on the assumed strength of host effects that

alter how microbes behave to one another, an FMT from one donor may either be generally effective or not.

A conceptual understanding for the conditions under which FMT may work or fail is needed. However, I do not think the article in its current form can be of much help with that. Many results are somewhat tautologous (if a host changes how microbes behave, then microbes behave differently, including to C diff, and FMT fails). The language confuses data and simulations. Oftentimes a section begins with a recap of empirical studies followed by results from simple simulations of microbiota timelines. These are then discussed in terms of "patients" in a confusing way (e.g. line 392: "Our simulation results show that for rCDI patients with low taxonomic diversity, FMT works equally well with different donors" --- Unless I missed it, no data went into these simulations).

Furthermore, the article is extremely long, but it is not clear what the key result is. In fact, the authors themselves seem unsure since even the abstract only vaguely hints at required "shifts [of] focus from specific taxa or functions to a systems level understanding of the human gut microbiota using network and ecological approaches".

I recommend a rework of the article with a better focus and clearer narrative. Perhaps a very direct path can be conceived of that starts with their simple ecosystem model and then systematically lists conditions under which FMT fails (e.g. when hosts change interactions, nothing is predictable; when seeming "competitors" of C diff also promote species that themselves promote C diff, such "network effects" can mean adding such a "competitor" to a probiotic cocktail may be a bad idea; ...).

More specific comments below.

line 51: "51 During FMT, fecal material from a carefully screened" - screened for what? More specificity here would be useful to summarize what the 'state of the art' is.

line 56: it is a bit strange to mix up both actual trials with speculative opinion pieces in a paragraph describing the current use of FMT.

line 61: what is the 'the fecal-oral route'

line 171: "species with universal population dynamics following the GLV model. " Why are there "global dynamics"? what is this simulating? Can you explain the underlying biology for this? Are your results conditional on these global dynamics (e.g. control results without this complexity).

line 183: "note that some species might become extinct during the process,..." How? Your ODEs do not allow this.

line 198: To simulate the donor's healthy gut microbial composition, we randomly assemble a local community from the species pool with the only condition that *C. difficile* cannot colonize. (Otherwise, this doesn't represent a healthy gut microbiota at all.) Can you provide any evidence for this? Healthy asymptomatic individuals can be carriers of *C. diff*.

line 276: To verify that the gut microbial dynamics is universal for rCDI patients and their healthy donor, ... --- You are not verifying this, you are simulating something.

line 298 (caption figure 4): We randomly choose 20 healthy donors... You did not choose health donors but simulated some ODEs. When you discuss several empirical studies at lengths in your paper, this language is confusing.

Response to Reviewer #1

To import concepts and approaches from ecology is an excellent idea, not totally new, but still mostly unexplored. Designing optimal cocktails is a very interesting idea, and to do this based on systems analysis is really promising. Theoretically speaking, the paper is very wise and perspectival. The question is feasibility and applicability, see my comments below, both minor and major.

We thank Reviewer #1 for her/his overall positive assessment of the general interest and potential of our framework. We share her/his belief that systems analysis is really promising for the design of optimal cocktails. Next we address each issue raised by her/him in order.

Figure 1, legend: "high taxonomic diversity", yes, this is true but the key point, according to the ecological approach of the paper, is high functional diversity.

We thank Reviewer #1 for pointing this out. We have revised that sentence (**page 23, lines 867-868**) as follows:

“In an initial healthy gut, the microbial community typically contains many different taxa and display very high taxonomic and functional diversity.”

"dynamics analysis" could be changed to "dynamical analysis"

We thank Reviewer #1 for this suggestion. We have replaced “dynamics analysis” by “dynamical analysis” throughout the revised manuscript.

In passing, it would be a lot easier for the reviewer to use line numbering....

Sure. We have added line numbers in the revised manuscript.

"When modeling a dynamical ecosystem, we first decide how complex the model needs to be so as to capture the phenomenon of interest." - this sentence is more like in textbook style, I suggest to remove this.

We thank Reviewer #1 for pointing this out. We have removed this sentence from the revised manuscript.

"Microbial interactions can be mediated by direct secretion of substances such as bacteriocins^{53,54}, competition for the same nutrient⁵⁵, or immune system modulation^{56,57,58}." - interactions between bacteria X and Y are really of composite nature, so the nature of the ultimate relationship is a black box. Also, to describe all these interactions experimentally is not feasible. This is why statistical associations are typically used for inferring positive and negative interactions. These can be used to parameterize dynamical models but the question is how reliable is this approach. You may elaborate these limitations.

We thank Reviewer #1 for this very insightful comment. We agree with her/him that experimentally characterizing all the inter-species interactions in complex microbial communities is rather challenging, if not impossible. Statistically associations are commonly used for inferring positive and negative “interactions”. But we emphasize that there are many fundamental limitations of this approach. We list some of the limitations here:

- 1) By definition, statistical associations are undirected (or bidirectional). But ecological interactions can be directed (or unidirectional): the impact of species X on the growth of species Y could be totally different from the impact of Y on X. Therefore, in general, statistical associations cannot capture ecological interactions.
- 2) Statistical associations can be strongly time- or state-dependent. Indeed, during the dynamical process of a community with well-defined ecological interactions, the statistical associations of species abundances can vary drastically over time. Over certain time window, the statistical association between the abundances of species X and Y can be positive. Over a different time window, the association can be negative or even zero (as clearly demonstrated in Fig.1 of Ref.[R1]).
- 3) In general, there is no simple relation between ecological interaction strengths and statistical correlations between species abundances (as shown in Fig.1 of Ref.[R2]).

Considering those limitations, we do not recommend the use of statistical associations in parameterizing dynamical models. The statistical associations can be easily calculated from the species abundance table, but they cannot be used to faithfully predict the temporal behavior of microbial communities. After all, associations/correlations are not causation.

Figure 2: the framework is nice and the inspiration from ecological literature is great. But only theoretically. Because of the above's, no one can parameterize these interactions today.

We thank Reviewer #1 for this very legitimate concern. The ecological network discussed in our paper is a directed, signed, and weighted graph, where nodes represent microbial taxa and edges represent direct ecological interactions between different taxa (e.g., parasitism, commensalism, mutualism, amensalism, or competition). Indeed, the ecological network of the human gut microbiota has not been mapped yet. As we mentioned in our paper, various methods have been developed to infer the ecological interactions of microbial communities from time-series [R1]-[R4] or steady-state data [R5]. Hence, currently, the major issue is not the lack of powerful inference methods, but poor data quality and small sample size. We are optimistic that, via community efforts, in the near future we can eventually collect high-quality data and successfully map the ecological network of human gut microbiota. Hence the theoretical ecological framework developed here can be eventually applied to solve real-world problems in the future. Moreover, we emphasize that our theoretical framework already made several predictions that are consistent with observations from real experiments (see **page 16** and **Figs.R1-R3** of this response letter).

Moreover, theoretical results should be tested experimentally. Artificial guts exist but their readiness for community-level tests should be briefly mentioned or discussed.

We thank Reviewer #1 for this very constructive comment. In the revised manuscript (**page 16, lines 590-592**), we briefly mention artificial guts and discuss their readiness for community-level tests as follows:

“Artificial guts (such as the gut-on-a-chip¹¹⁰ and the HuMiX¹¹¹ system) would also be intriguing to test our predictions, though an important challenge still lies in further increasing their high-throughput analyses capacity¹¹².”

Figure 2 legend: change "Principle" to "Principal"

We thank Reviewer #1 for pointing this out. We have fixed this typo in the revised manuscript.

I am not sure how the steady state condition can be understood. The mammal gut microbiome is a highly dynamical system, sensitive to environmental effects and actual food consumption. To what extent can we speak about steady state or any kind of balance? What is the time-scale for this?

We thank Reviewer #1 for this very insightful comment. We apologize for not making this point clear in the previous version of our manuscript.

Indeed, the mammal gut microbiome is a highly dynamical ecosystem. But we emphasize that, though human gut microbiota can be quickly and profoundly altered by common day-to-day human actions and experiences [R6], several previous studies show that normally human gut microbiota remains stable for months, and possibly even years [R7]-[R9]. That is, without drastic diet change or repeated antibiotic administrations, the gut microbiome of healthy adults remains close to a steady state and tends to return to this state even after it is perturbed. From a dynamical systems viewpoint, this means that the healthy human gut microbiota has an asymptotically stable equilibrium state. It is this stability property that allows the human gut microbiota to withstand and recover from (mild) perturbations. Below we list a few observations in the literature that support this point:

- 1) It has been shown that short-term consumption of diets composed entirely of animal products shifts the community structure of human gut microbiota, but subjects' gut microbiota returned to their original structure only two days after the animal-based diet ended [R10].
- 2) Preparing for a colonoscopy requires clearing the bowel with fasting, a laxative drink and, in some cases, an enema. While such preparation can alter the gut microbiota, it has been found the human gut microbiota returns closely to its original structure in about 2 to 4 weeks [R11].
- 3) Upon closer inspection of the longitudinal gut microbiome data shown in Figure 3A of [R12], it has been found that the trajectories always returned to the same steady region in a few days [R13].

To demonstrate the stability of the human gut microbiota, we project the nearly daily abundance profiles from male and female subjects in the Moving Pictures Study (MPS) [R9] onto the principal components (PC) computed from 353 stool microbiome samples of the Human Microbiome Project (HMP) **Error! Reference source not found.** As shown in **Fig.R4**, over a typical 10 to 20-day window, the abundance profile of each subject stays in the same region of the PC space. This behavior suggests the presence of a stable state that attracts and keeps the microbiota in that region. Large excursions away from that region occur in the male subject, possibly due to large external perturbations such as drastic diet changes. But, crucially, the gut microbiota quickly (in a few days) returns to the original region. This ability to recover in response to perturbations is only possible thanks to the asymptotical stability of the human gut microbiota. That's why in our framework we assume that normally the human gut microbiota is in a stable steady-state.

It is unclear what is meant by strong universality of gut microbiome samples. There is huge variability exactly in the HMP database. Some systemic outputs can be more general.

We apologize for not making this point clearly in the previous version of our manuscript. Since other reviewers also raise similar concerns about the universality of gut microbial dynamics, we address them all together at the end of the response letter (see **page 18**, as well as **Fig.R5**).

"To find a compatible donor for a given recipient, one may consider a naive approach: compare their taxonomic profiles, and calculate the fractions of donor-specific taxa (denoted as f_d), recipient-specific taxa (f_r), and common taxa ($f_c=1-f_d-f_r$)." - this approach is a good one but it dramatically reduces the holistic, systemic information provided by the network approach.

We fully agree with Reviewer #1 that the analysis based on the triple (f_d, f_r, f_c) is problematic. Our simulation results (Fig.5 in main text) clearly demonstrate that this method cannot distinguish responders from non-responders. We also applied this method to analyze a real dataset [R15], finding again that it cannot distinguish responders from non-responders (**Fig.R2f-j**).

The difference between cord blood banking and the idea of fecal sample banking is that animal development is a much more conservative process, while the development (assembly) of the microbiota is very much exposed to environmental effects. I would trust more in variability and selection (trial and error) between different treatments.

We thank Reviewer #1 for this insightful comment. We agree that the idea of stool banking for future autologous FMT is just **conceptually** similar to the purpose of cord blood banking. Our simulations demonstrate that autologous FMT works extremely well. Of course, well-designed clinical trials are required to further test this idea.

To potentially improve the link to ecological literature, let me suggest some further references:

FMT sounds more like an invasion meltdown situation, see
<https://esajournals.onlinelibrary.wiley.com/doi/abs/10.1890/11-0050.1>

For the relationship between invasion and community health, see the famous example of Mnemiopsis invasion of the Black Sea (disturbed community was more invadable):
<https://onlinelibrary.wiley.com/doi/abs/10.1046/j.1365-2419.1998.00080.x>

For identifying critically important elements in complex dynamical ecosystems, see:
<https://www.nature.com/articles/srep15920>

We thank Reviewer #1 very much for pointing out those important references. We have cited them appropriately in the revised manuscript.

The suggested modelling approach remains a theoretical exercise for many species. A possibly more realistic approach would be to parameterize dynamical models and simulate them for small network modules of 3-4-5 species. In this case, interactions could even be measured

experimentally and then realistic models could be built, following the same kind of ecological logic.

We thank Reviewer #1 for this very constructive comment. Since other reviewers also raised very similar concern about testing our theoretical results experimentally or/and applying our framework to real microbial communities, we addressed this comment all together in the end of this response letter (see **Page 16**, as well as **Figs.R1-R3**).

Finally, we thank Reviewer #1 again for her/his very insightful and constructive comments. We hope our responses above have addressed those very legitimate issues/concerns in a satisfactory manner.

Response to Reviewer #2

The manuscript by Xiao et al. describes an ecological framework to study the efficacy of FMT to treat rCDI. This framework utilizes changes in bacterial taxa pre- and post-FMT (and FMT donor taxa) to propose bacteria that are important for inhibiting *C. difficile* growth. This ecological framework is interesting and potentially a very useful tool to design personalized synthetic microbial communities to treat diseases. However, I'm not sure how applicable this work will be in the absence of functional data (especially in the case of CDI). Moreover, some of the claims made in this paper for rCDI do not align with what we know of the mechanisms of FMT to treat rCDI. Finally, the manuscript as it exists in its current form relies on simulated data, and needs to be validated with in vivo and in vitro data.

We thank Reviewer #2 for her/his positive assessment on the general interest and potential application of our ecological framework proposed in this manuscript. We next address each of the reviewer's concerns in order.

Comments:

- Lines 67-68: The authors write "In a healthy gut microbiota, *C. difficile* is "out-competed" by hundreds of strains of bacteria that are normally present (Fig.1a)." However, this implies the mechanism of FMT is competition for nutrients (which isn't precisely correct based on our current understanding of the mechanism of FMT for rCDI). Reword to "In a healthy gut microbiota, *C. difficile* is unable to colonize the gut in the presence of the hundreds of strains of bacteria that are normally present (Fig. 1a)." Or something similar.

We thank Reviewer #2 for this insightful comment. We fully agree with him/her that "out-compete" could be quite misleading. We have revised that sentence following Reviewer #2's suggestion (see main text, **page 3, lines 69-70**).

- Line 70 (and throughout the manuscript, including Fig 1): change "flourish" to "grow"

We thank Reviewer #2 for pointing this out. We have replaced "flourish" by "grow" throughout the manuscript.

- Line 72 (and throughout the manuscript): "flora" is an outdated term. "Microbiota" is the preferred term.

We thank Reviewer #2 for pointing this out. We have replaced "flora" by "microbiota" throughout the manuscript.

- Line 73: Change "and recurs in" to "and CDI recurs in"

Done. Thanks.

- Lines 93-95: The authors state "Yet, we still don't understand why FMT works so well for rCDI and why ~20% of rCDI patients relapse after FMT. In fact, very little is known about the

underlying mechanism of FMT and its long-term effects from the ecological perspective.” I disagree. There have been several studies that demonstrate the mechanism of FMT to treat rCDI. I would reword to say that we have only recently started to understand the mechanisms of FMT to treat rCDI.

We thank Reviewer #2 for this very constructive comment. We have revised that sentence accordingly and cited several relevant papers that demonstrated the mechanisms of FMT in treating rCDI (see main text, **page 3, line 81**).

- Line 94: Missing a key reference that describes the mechanism of FMT to treat CDI. Add the following reference: McDonald JAK et al. (2018) Inhibiting Growth of *Clostridioides difficile* by Restoring Valerate, Produced by the Intestinal Microbiota. *Gastroenterology*. 155(5):1495-1507.

We thank Reviewer #2 for pointing out this very important reference. We have cited it in the revised manuscript.

- Lines 97-98: The authors state “Why does FMT work better for recurrent CDI than for primary CDI?” FMT is used for rCDI more than primary CDI because it is not a first line treatment for primary CDI (antibiotics are the first line treatment). Has anyone ever compared the efficacy of FMT to treat primary CDI vs rCDI? If so, a reference is needed here.

We thank Reviewer #2 for this very insightful comment. We could not find any published work **directly** comparing the efficacy of FMT in treating **primary CDI vs. rCDI** in the same clinical study. We did notice a paper [R16], where the authors undertook a proof-of-concept trial (ClinicalTrials.gov number, NCT02301000) to evaluate the use of FMT as treatment for primary CDI. In particular, they found that the overall response to treatment (full primary or secondary response) was achieved in 7 (of 9) patients in the **FMT group** (78%; 95% CI, 40 to 97), as compared with 5 (of 11) in the **metronidazole group** (45%; 95% CI, 17 to 77) (p-value = 0.20). Although the results suggest that FMT may be an alternative to antibiotic therapy in primary CDI, the response rate (78%) is lower than the rate of FMT in treating rCDI (~92%) based on a meta-analysis of 37 studies [R17]. Of course, this proof-of-concept trial is very small. Hopefully, in the future we can get more accurate response rate of FMT in treating primary CDI.

Using our ecological framework, we actually demonstrated that the FMT efficacy is negatively correlated with the taxonomic diversity of the recipient’s pre-FMT microbiota. This finding is partially consistent with the real FMT outcomes of a clinical trial [R15]. We actually reanalyzed the data in [R15], finding that responders typically have lower median taxonomic diversity (in terms of species richness, Shannon index, and Simpson index) than non-responders (**Fig.R2c-e**). Yet, this difference is not statistically significant (very likely due to the imbalance of samples size: responders n=71 vs. non-responders n=17). More studies are certainly needed to fully understand the relationship between the taxonomic diversity of the recipient’s pre-FMT microbiota and the FMT efficacy. Since primary CDI patients typically have less dysbiotic and more diverse gut microbiota than that of rCDI patients [R18], our simulation results suggest that FMT should have lower efficacy in treating primary CDI than rCDI.

To be more precise, in the revised manuscript (**page 3, line 86**), we have changed the sentence to be “*Does FMT work equally well in treating primary and recurrent CDI?*”

- I cannot comment on the mathematical equations described in this manuscript, as this is outside of my field (lines 117-243 and lines 421-432).

We thank Reviewer #2 for this comment. All the mathematical equations described in our paper are based on classical population dynamics models in community ecology. Analytical calculations are based on dynamical systems theory and matrix theory. Details can be found in the SI Sec.2.

- Lines 233-235: When a gut microbiota is disrupted by antibiotics, it not only results in killing of members of a healthy gut microbiota, but also the bloom of members of the healthy gut microbiota that are resistant to the antibiotic. Does your ecological framework take this into account, or is it only presence/absence?

Yes. Our ecological framework takes this into account. In principle, as long as the antibiotic removes species that effectively inhibit the growth of certain commensal species, we should anticipate the bloom of this commensal species after antibiotic administration. In the simple simulation shown in Fig.2g, seven species survived after antibiotic administration, and four of them (including *C. difficile*) displayed increased abundance. To better illustrate this point, see **Fig.R6** in the end of this response letter, where we highlight the abundance time series of the four species in thick lines. In the revised manuscript (**page 25, lines 907-908**), we also briefly mention this point as follows:

“Note that seven species can survive after the hypothetical antibiotic administration, and four of them (including *C. difficile*) display increased abundance.”

- Lines 257-259: The authors state “For example, in treating rCDI with FMT, an extreme scenario could be that the same set of microbes that effectively inhibit the growth of *C. difficile* in the donor’s gut might promote the growth of *C. difficile* in the recipient’s gut.” This is very unlikely. FMT banks that provide stool from the same donor to multiple rCDI patients have not demonstrated this. In fact, FMT has never been shown to promote CDI (FMT is very effective for treating rCDI, regardless of the FMT donor’s gut microbiota composition).

We thank Reviewer #2 for this insightful comment. Indeed, this will be very unlikely. In the revised manuscript, we have removed that sentence to avoid being misleading.

- Line 269: What do the authors mean by the term “universality”?

Here, “universality of their gut microbial dynamics” means that the dynamical rules (which can be parameterized in terms of intrinsic species growth rates and inter-species interactions) are host-independent. In other words, we can use the same population dynamics model (i.e., the same set of ordinary differential equations) to predict the temporal behavior of the microbial ecosystem of different subjects. Different subjects just differ by the collections of the species colonized in their guts. But their common species interact in a host-independent way. We apologize for not making this point clearly in the previous version of our manuscript. Since other reviewers also raise similar concerns, we address them all together at the end of the response letter (see **page 18**, as well as **Fig.R5**).

- Lines 271-275: The gut microbiota of rCDI patients are variable because the patient cohort of quite heterogeneous: many different kinds of patients (on a variety of diets/medications), all receiving a wide variety of antibiotics that result in a first episode of CDI, then have varying recovery following antibiotics before they develop CDI, then who go on to receive a broad spectrum antibiotic to treat CDI, then who have varying recovery responses before developing rCDI, who then get more broad spectrum antibiotics before receiving FMT. In addition to the varying starting composition of their gut microbiota, they have different histories. So it's not unexpected that rCDI patients do not have similar community dynamics.

We thank Reviewer #2 for this very insightful comment. Indeed, rCDI patients have not only very different starting gut microbial compositions but also have received very different antibiotic administration and recovery histories. We have added this very insightful point to the revised manuscript (**page 7, lines 222-229**) to better explain why the universality of the gut microbial dynamics of rCDI patients was not observed in their pre-FMT gut microbiota.

- Lines 343-345: The authors state “It is reasonable to assume that primary CDI patients have less dysbiotic and more diverse gut microbiota than that of rCDI patients.” This has been demonstrated in a published study (add this reference): Allegretti JR, et al. (2016) Recurrent Clostridium difficile infection associates with distinct bile acid and microbiome profiles. Aliment Pharmacol Ther. 43(11):1142-53

We thank Reviewer #2 very much for pointing out this very important reference, which strongly supports the assumption made in our framework. We have cited this paper in the revised manuscript (**page 8, lines 277-279**).

- Lines 359-361: Another (more probable) interpretation is that when you add more species, either they have no effect on C. difficile growth, or they have functional redundancy (so adding them makes no difference). Or, if adding more species reduces FMT efficacy, then it could be that a species that does not affect C. difficile growth is inhibiting the growth of a species that can inhibit C. difficile growth.

We thank Reviewer #2 for this insightful comment. Indeed, the network effort could have many different scenarios. In the revised manuscript, we have mentioned those scenarios suggested by Reviewer #2 (see **page 9, lines 292-294**).

- Lines 368-370: A more probable explanation is that the donor strains could not engraft because of competition between the recipient's existing bacteria and the bacteria in the FMT. Based on what we know about the mechanism of FMT to treat CDI I do not believe bacteria in the FMT donor would promote C. difficile growth in a CDI patient.

We thank Reviewer #2 for this very insightful comment. In the revised manuscript, we have replaced that sentence by the following one (see **page 9, lines 301-303**):

“Probably the donor microbiota cannot easily engraft because of competition between the recipient's existing species and the species in the FMT.”

- Lines 393-395: But rCDI patients don't have high diversity of their gut microbiota. How do you define high diversity in these patients? And can you confirm that this patient was colonized with

vegetative *C. difficile* (not spores) at the time the sample was collected? Do you have data to compare the levels of *C. difficile* in patients with low diversity to levels found in patients with high diversity (not 16S rRNA gene sequencing data, but qPCR levels or plate counts)?

We thank Reviewer #2 for this very insightful comment. We apologize for this very misleading sentence: “However, for rCDI patients with higher taxonomic diversity certain donors work much better than others (Fig.5b,c), implying a pronounced donor-recipient compatibility issue”. We should have made it clear that we were talking about **simulation results**. In our simulations, we can simulate microbial communities that mimic the gut microbiota of rCDI or primary CDI “patients” by tuning the species richness (so that the pre-FMT microbiota of rCDI “patients” has much lower species richness than that of primary CDI “patients”). For rCDI “patients”, we can use the very same idea to simulate microbial communities with different taxonomic diversity levels (by simply tuning species richness) so that some rCDI “patients” will have **relatively** higher taxonomic diversity than other rCDI “patients”.

To avoid any potential confusion, we have revised that sentence as follows (see **page 9, lines 313-315**):

“Moreover, our simulations suggest that the FMT efficacy will decrease with increasing taxonomic diversity in rCDI patients (Fig.4d-f), implying a pronounced *donor-recipient compatibility* issue.”

• Lines 477-480: I am not convinced by this argument based on what we know about the mechanism of FMT to treat rCDI. We cannot reply on 16S data alone, but we must also look at changes in the function of the community. This would greatly strengthen this model.

We thank Reviewer #2 for this very insightful comment. We fully agree with her/him that explicitly considering functional profiles will greatly strengthen our modeling framework. From the mathematical modeling perspective, it is always preferred to start with a minimal model to facilitate the parameterizing procedure, which is already a daunting task even if we deal with taxonomic profiles (i.e., species abundances) only. Since we are developing the very first ecological modelling framework to understand the efficacy of FMT, we chose to work with the classical Generalized Lotka-Volterra population dynamics model. Although this model is very simple and phenomenological, it accommodates various possible inter-species interactions found in community ecology (e.g., parasitism, commensalism, mutualism, amensalism, or competition).

We fully agree with Reviewer #2 that additional efforts should be dedicated to combine both taxonomic and functional data to obtain a more comprehensive modeling framework. We just feel this is beyond the scope of the current research and deserves further study. In the revised manuscript, we have explicitly mentioned the limitation of our current modeling framework, and mentioned this potential future research direction in the discussion section (**page 16, lines 576-584**).

• Lines 487-490: But FMT works well to treat rCDI, regardless of the donor and how the FMT was prepared. This suggests restoration of a microbial function (that has redundancy across species) is responsible for treating CDI. This is supported by what we understand about the mechanism of FMT to treat rCDI.

We thank Reviewer #2 for this very insightful comment. We fully agree with her/him that restoration of microbial functions (such as secondary bile acid metabolism) is responsible for the

success of FMT in treating CDI. By “magic bullet” we just meant a unique probiotic cocktail that works for all rCDI patients. Our simulation results suggest that such a generic probiotic cocktail does not exist. We didn’t mean there will be no super donor whose fecal material works for all rCDI patients through FMT. (Actually, our simulation result shown in Fig.5 indicates the existence of such super donors.) To design a truly personalized probiotic cocktail that works for a specific patient, we have to take into account the patient’s diseased microbiota. But, as pointed out by Reviewer #2, those personalized cocktails (though consisting of different combinations of species) could restore the microbial function in a generic way (thanks to the functional redundancy of those microbial species).

To avoid any potential confusion, in the revised manuscript (**page 11, line 388**), we revised the sentence as follows:

“*Second*, a “magic bullet” (i.e., a unique combination of microbial species) that works for all patients very likely doesn’t exist.”

Moreover, we added the following sentences in the discussion section (**page 16, lines 580-584**) to emphasize the importance of functional data:

“Moreover, such an integrative modeling/analysis of taxonomic and functional data will enable us to better design personalized probiotic cocktails. Indeed, though consisting of different combinations of species, those personalized probiotic cocktails could restore the microbial functions (such as secondary bile acid metabolism) in a generic way, thanks to the functional redundancy of microbial species.”

• Lines 511-513: This is too simplistic, as some microbial processes require many species to work together to carry out a function. You cannot simply look at which species are directly inhibiting *C. difficile*, as you may need additional species to carry out some steps first.

We fully agree with Reviewer #2 that we cannot simply look at which species are **directly** inhibiting *C. difficile*. (Our simulation results shown in Fig.6 and real data analysis shown in Fig.9 clearly demonstrated that this strategy will not work.) That’s exactly why our tentative cocktail composes of all the **effective** inhibitors, regardless of **direct** or **indirect** inhibition. Those indirect inhibitors have a negative impact on the growth of *C. difficile* through other “mediator” species.

To make this point clearer, in the revised manuscript (**page 11, line 404**), we added the following sentence:

“Note that effective inhibitors include both direct and indirect inhibitors.”

• Much of this work relies on simulated data sets. These methods need validation on published data sets.

We thank Reviewer #2 for this very constructive comment. Since other reviewers raised very similar concerns, we addressed them all together in the end of this response letter (see **page 16**, as well as **Figs.R1-R3**).

Finally, we thank Reviewer #2 again for reviewing our paper. We hope our responses above have addressed those very legitimate concerns in a satisfactory manner.

Response to Reviewer #3

I have now read "An Ecological Framework to Understand the Efficacy of Fecal Microbiota Transplantation" by Liu and colleagues. They present computer simulations of microbial ecosystem dynamics using several established ecological models. The authors use these equations to study how the rise of *C. difficile* (following a simulated dose of antibiotics) may be remedied with a simulated fecal microbiota transplantation.

The gut microbiota community is modeled as sets of differential equations representing the growth rates of individual species. *C. difficile* is one of these species. The authors investigate under what conditions the abundance of *C. diff* is reduced and a pre-antibiotic community is restored. They also consider the potential for the between-species interactions to differ between hosts. The authors then simulate the development of species abundances over time. Simulating ODEs can be educational, and the authors here show that depending on the assumed strength of host effects that alter how microbes behave to one another, an FMT from one donor may either be generally effective or not.

We thank Reviewer #3 for her/his comprehensive review of our framework. We next address each of her/his concerns in order.

A conceptual understanding for the conditions under which FMT may work or fail is needed. However, I do not think the article in its current form can be of much help with that. Many results are somewhat tautologous (if a host changes how microbes behave, then microbes behave differently, including to *C. diff*, and FMT fails). The language confuses data and simulations. Oftentimes a section begins with a recap of empirical studies followed by results from simple simulations of microbiota timelines. These are then discussed in terms of "patients" in a confusing way (e.g. line 392: "Our simulation results show that for rCDI patients with low taxonomic diversity, FMT works equally well with different donors" --- Unless I missed it, no data went into these simulations).

We thank Reviewer #3 for pointing this out. We apologize for the misleading writing style of our paper. In the revised version, we have paid special attention to this issue and tried our best to clarify what is known from empirical studies in the literature and what is predicted by our simulation framework. As for using real data to validate our simulations, since other reviewers raised very similar concerns, we addressed them all together at the end of this response letter (see **page 16**, as well as **Figs.R1-R3**).

Furthermore, the article is extremely long, but it is not clear what the key result is. In fact, the authors themselves seem unsure since even the abstract only vaguely hints at required "shifts [of] focus from specific taxa or functions to a systems level understanding of the human gut microbiota using network and ecological approaches".

We thank Reviewer #3 for this critical comment. We apologize for not making our key results clear enough. In this paper —by combining community ecology theory, mathematical modeling, dynamics analysis, and network science— we built a framework to characterize the principles underlying the effectivity of FMT, additionally showing how this framework can be used to design better probiotic cocktails. Specifically, we first revealed and characterized three key factors that determine the success of FMT (i.e., universal microbial dynamics, low taxonomic

diversity of the pre-FMT microbiota, and donor-recipient compatibility). Second, we develop a network-based method to design probiotic cocktails containing only the effective components of FMT, offering new insights for the rational design of microbiota-targeted therapeutics. Finally, in the revised version, we analyzed several real datasets to validate our simulation framework. In particular, we demonstrated the rational design of probiotic cocktails in decolonizing *C. difficile* using an ecological network inferred from mouse experiments.

In the revised manuscript (**page 1, lines 19-24**), we have revised the Abstract accordingly to emphasize our key results as follows:

“Here, we present an ecological framework to understand the efficacy of FMT in treating conditions associated with a disrupted gut microbiome, using recurrent *Clostridioides difficile* infection as a prototype disease. This ecological framework predicts several key factors that determine the efficacy of FMT. Moreover, it offers an efficient algorithm for the rational design of probiotic cocktails to decolonize certain pathogens. We analyze both *in vitro* and *in vivo* datasets to further validate our theoretical predictions.”

I recommend a rework of the article with a better focus and clearer narrative. Perhaps a very direct path can be conceived of that starts with their simple ecosystem model and then systematically lists conditions under which FMT fails (e.g. when hosts change interactions, nothing is predictable; when seeming "competitors" of C diff also promote species that themselves promote C diff, such "network effects" can mean adding such a "competitor" to a probiotic cocktail may be a bad idea; ...).

We thank Reviewer #3 for this very constructive comment. We have heavily revised our manuscript. Now it has the following structure:

1. Introduction

2. Theoretical Results

2.1 Ecological Modeling Framework

2.2 Reveal Key Factors that Determine the Success of FMT

2.2.1 Impact of host-dependent microbial dynamics on FMT efficacy

2.2.2 Impact of taxonomic diversity of recipient's pre-FMT microbiota on FMT efficacy

2.2.3 Impact of donor-recipient compatibility on FMT efficacy

2.3 Towards a rational design of probiotic cocktails to treat rCDI

3. Real Data Analysis

3.1 Network effect in real microbial communities

3.2 Taxonomic diversity of pre-FMT microbiota of responders and non-responders

3.3 The donor-recipient compatibility issue in a clinical trial

3.4 Rational design of probiotic cocktails for a real microbial community

4. Discussion

More specific comments below.

line 51: "51 During FMT, fecal material from a carefully screened" - screened for what? More specificity here would be useful to summarize what the 'state of the art' is.

We thank Reviewer #3 for this comment. In the revised manuscript, we added the following sentence to summarize the donor exclusion criteria (see **page 2, lines 48-51**):

“Both absolute and relative contraindications have been proposed for donor screening^{16,21,22}. Absolute contraindications include the risk of infectious agent, GI comorbidities, etc.; while relative contraindications include history of major GI surgery, metabolic syndrome, systemic autoimmunity, etc.”

line 56: it is a bit strange to mix up both actual trials with speculative opinion pieces in a paragraph describing the current use of FMT.

We thank Reviewer #3 for pointing this out. To avoid confusion, in the revised manuscript, we have revised that sentence as follows (see **page 2, lines 51-58**):

“FMT has been successfully used in the treatment of recurrent *Clostridioides difficile* infection (rCDI)^{10,19,23-28}. Numerous case reports and cohort studies have described the use of FMT in patients with inflammatory bowel disease²⁹⁻³². FMT has also been experimentally used to treat many other GI diseases such as irritable bowel syndrome (IBS)³³⁻³⁵ and allergic colitis³⁶, as well as a variety of challenging non-GI disorders such as autism³⁷, obesity³⁸, multiple sclerosis³⁹, hepatic encephalopathy⁴⁰, and Parkinson’s disease⁴¹. Larger multicenter studies and standardized double blinded randomized clinical trials are certainly needed to fully evaluate the efficacy of FMT in treating those diseases beyond rCDI.”

line 61: what is the 'the fecal-oral route'

We thank Reviewer #3 for this question. *C. difficile* bacteria and their spores could be found in feces. People can get infected if they touch surfaces contaminated with feces, and then touch their mouth. Healthcare workers can spread the bacteria to their patients if their hands are contaminated.

line 171: "species with universal population dynamics following the GLV model. " Why are there "global dynamics"? what is this simulating? Can you explain the underlying biology for this? Are your results conditional on these global dynamics (e.g. control results without this complexity).

We apologize for not making this point clearly in the previous version of our manuscript. Here the “universal” population dynamics means that different local communities (i.e., the gut microbiota of different subjects) share the same ecological rules (which can be parameterized by the same set of intrinsic species growth rates, and inter-species interactions), and hence can be described by the same set of ordinary differential equations. Different subjects just differ by the collection of species present in their guts. Their common species interact in a host-independent way.

Since other reviewers also raise similar concerns about “universal dynamics of human gut microbiome”, we address them all together at the end of the response letter (see **page 18**, as well as **Fig.R5**).

line 183: "note that some species might become extinct during the process,..." How? Your ODEs do not allow this.

We thank Reviewer #3 for this comment. For a local community evolving from any initial state, some species present at the beginning might become extinct during the time evolution (see **Fig.R7** for an example). For the details of our simulations, please see Sec. 1 of Supplementary Information.

line 198: To simulate the donor's healthy gut microbial composition, we randomly assemble a local community from the species pool with the only condition that *C. difficile* cannot colonize. (Otherwise, this doesn't represent a healthy gut microbiota at all.) Can you provide any evidence for this? Healthy asymptomatic individuals can be carriers of *C. diff*.

We thank Reviewer #3 for this very insightful comment. Indeed, there are healthy asymptomatic individuals who can be carriers of *C. difficile*. In our modeling framework, those asymptomatic carriers were not considered. To make this point clearer, in the revised manuscript we added the following sentence (see **page 5, lines 175-178**).

“This is of course a simplified modeling approach. In reality, there are asymptomatic carriers⁸², i.e., with presence of toxicogenic *C. difficile* in their colon but no symptoms of CDI. This is not considered as “healthy” in our modeling framework.”

line 276: To verify that the gut microbial dynamics is universal for rCDI patients and their healthy donor,... --- You are not verifying this, you are simulating something.

We thank Reviewer #3 for pointing this out. In the revised manuscript, we have removed that sentence to avoid confusion.

line 298 (caption figure 4): We randomly choose 20 healthy donors... You did not choose health donors but simulated some ODEs. When you discuss several empirical studies at lengths in your paper, this language is confusing.

We apologize again for the misleading writing style. We have heavily revised our manuscript to resolve this issue. In particular, now we have two separate sections “**2. Theoretical Results**” and “**3. Real data analysis**”. And we added the following paragraph (**page 3, lines 101-107**) to introduce Section 2 and emphasize that results in this Section are based on simulating the dynamics of certain microbial communities that mimic the gut microbiota of patients and donors.

“In this section, we first propose an ecological modeling framework to simulate the FMT process. This modeling framework then enables us to predict several key factors that determine the efficacy of FMT. Moreover, it helps us develop an efficient algorithm for the rational design of probiotic cocktails to decolonize a pathogenic species. All the results presented in this section are based on FMT simulations, unless otherwise stated. Moreover, both “donors” and “recipients” discussed in the FMT simulations just represent the “hosts” of different simulated microbial communities. They should not be confused with real human subjects in clinical studies.”

Finally, we thank Reviewer #3 again for her/his very constructive comments and suggestions. We hope our responses above have addressed those very legitimate issues/concerns in a satisfactory manner.

Responses to common concerns from reviewers

1. The modelling approach remains a theoretical exercise and needs validation on published datasets.

To directly address this concern, we analyzed both *in vitro* and *in vivo* microbiome datasets to test our theoretical predictions. First, we demonstrated the ubiquitous network effect in real microbial communities. Then, we compared the taxonomic diversity of pre-FMT microbiota of responders and non-responders in a clinical trial of FMT. We also demonstrated that the naive approach of comparing the microbial compositions of donors and recipients will not resolve the donor-recipient compatibility issue. Finally, we numerically demonstrated the power of probiotic cocktails designed by our algorithm to decolonize *C. difficile* from a real microbial community.

(1) Demonstrate the ubiquitous network effect using both *in vitro* and *in vivo* data.

In our modeling framework, we proposed the concept of “network effect”. For example, those species that directly inhibit the growth of *C. difficile* might also indirectly promote the growth of *C. difficile* through other “mediator” species. The net or effective impact of a species on the growth of another species is hence largely context dependent. Specifically, if we compare the direct impact with the net impact, there are three cases: (i) *normal*: the direct and net impacts share the same sign; (ii) *bridging*: the direct impact is zero while the net impact is not; and (iii) *counter-intuitive*: the direct and net impacts have opposite signs. To directly demonstrate the presence of network effect, we analyzed two synthetic microbial communities constructed *in vitro* and *in vivo*, respectively.

Figure R1a illustrates an ecological network inferred from the abundance data of a synthetic soil microbial community of eight species [R19]. This data set consists of microbiome samples of all 8 solos, 28 duos, 56 trios, all eight septets, and one octet. The ecological network shown in Fig.R1a was inferred by the temporal data of solos and duos based on the assumption that the bacterial community follows the generalized Lotka–Volterra (GLV) dynamics. Fig.R1b focuses on the analysis of 56 trios. The upper row in Fig.R1b showed the local interaction matrix and corresponding contribution matrix of a subcommunity composed of three species *Ea*, *Pa*, *Pch*. We found that there exist two counter-intuitive cases (two red boxes in Fig.R1b): the sign of direct impact of species *Pch* on *Ea* (or *Ea* on *Pa*) is different from the sign of net impact. Interestingly, the counter-intuitive cases appear in almost all the 56 trios.

Figure R1c illustrates the interaction matrix inferred from the mouse experiments of antibiotic-mediated CDI [R3]. The experiments consisted of three populations of mice: (i) The first population received spores of *C. difficile*, and was used to determine the susceptibility of the native microbiota to invasion by the pathogen. (ii) The second population received a single dose of clindamycin to assess the effect of the antibiotic alone. (iii) The third population received a single dose of clindamycin and, on the following day, was inoculated with *C. difficile* spores. GLV model with the additional of external perturbations (i.e., antibiotic) was used to infer the ecological dynamics (e.g., intrinsic growth rates, inter-species interactions, etc.). From the inferred interaction matrix (Fig.R1c), we calculated the contribution matrix (Fig.R1d), where zero rows and columns represent species that cannot coexist with other species in equilibrium. Counter-intuitive effects between species were highlighted by red boxes in Fig.R1d.

Those results indicate that network effect is ubiquitous in microbial communities, illustrating the necessity of using our ecological modelling framework to understand the efficacy of FMT.

(2) Clinical evidence that partially supports our simulation result that FMT efficacy generally decreases with increasing taxonomic diversity of the pre-FMT microbiota.

Our modeling framework predicted that FMT efficacy is negatively correlated with the taxonomic diversity of the recipient's pre-FMT microbiota. To test our prediction, we analyzed real FMT data from a clinical trial [R15], where in total 106 rCDI patients were treated with encapsulated donor material for FMT (cap-FMT). Fig.R2a shows the sequenced fecal samples from 7 healthy donors and 88 rCDI patients at different time points: pre-FMT, 2-6 days post-FMT, weeks (7-20 days) post-FMT, months (21-60 days) post-FMT, and long-term (> 60 days). (Fecal materials from some patients were not available for sequencing in the trial.) Fig.R2b shows the PCoA plot of those samples, from which it is hard to distinguish responders from non-responders.

Interestingly, we found that those non-responders of cap-FMT tend to have higher median taxonomic diversity than that of responders (Fig.R2c-e). This clinical evidence partially supports our simulation result that FMT efficacy generally decreases with increasing taxonomic diversity of the pre-FMT microbiota. Note that the difference is not statistically significant. We anticipate that this might be due to the imbalance between the sample sizes of the responders ($n = 71$) and the non-responders ($n = 17$). Further clinical studies are definitely needed to validate our theoretical prediction.

(3) Demonstrate the donor-recipient compatibility issue using clinical data.

To demonstrate the donor-recipient compatibility issue using real data, we analyzed real FMT data from the clinical trial of cap-FMT mentioned above [R15], where one donor's fecal material was transplanted into many different recipients. As clearly shown in Fig.R2a, for almost all of the donors, most recipients responded to the cap-FMT, but a few recipients did not. For each (donor, recipient) pair, we further calculated the fractions of donor-specific taxa (f_d), recipient-specific taxa (f_r), and common taxa ($f_c = 1 - f_d - f_r$) at different time points. We found that it is impossible to distinguish responders and non-responders in the ternary plot (Fig.R2f-j). This result is consistent with our simulation result shown in the main text Fig.5d-f.

(4) Design personalized probiotic cocktails for a microbial community *in vivo*.

In Ref [R4], the ecological network involving the so-called GnotoComplex microflora (a mixture of human commensal bacterial type strains) and *C. difficile* was inferred from mouse data. In particular, germ-free mice were first pre-colonized with the GnotoComplex microflora and the commensal microbiota were allowed to establish for 28 days. Then, mice were infected with *C. difficile* spores and monitored for an additional 28 days. From the ecological network (Fig.R3a1), we notice that species-4 (*C. scindens*) and species-13 (*R. hominis*) can directly inhibit the growth of *C. difficile*; while species-1 (*C. hiranonis*), species-3 (*P. mirabilis*), species-5 (*R. obeum*), species-7 (*B. ovatus*), and species-12 (*K. oxytoca*) can indirectly inhibit the growth of *C. difficile* through some mediating species. Based on this ecological network and the disrupted microbiota, we can design probiotic cocktails to effectively decolonize *C. difficile*.

In Fig.R3a1-a3, we show the initial community composed of all the 14 species in the network, a disrupted microbiota due to hypothetical antibiotic administration, and the restored community after the administration of a particular probiotic cocktail, respectively. Fig.R3b demonstrates the efficacy of various probiotic cocktails. The optimal probiotic cocktail R_{global} is designed based on the global ecological network (Fig.R3a1) and the specific disrupted microbiota (Fig.R3a2). This cocktail R_{global} contains two direct inhibitors of *C. difficile* (i.e., species-4: *C. scindens* and species-13: *R. hominis*), and two indirect inhibitors of *C. difficile* (i.e., species-5: *R. obeum* and species-12: *K. oxytoca*). As shown in Fig.R3b (green curve), this cocktail R_{global} can strongly suppress the abundance of *C. difficile*. We also designed several cocktails based on the n -step

ego-networks of *C. difficile* (with $n = 1,2,3$), which just contains those species that are n -step away from *C. difficile* in the original network (Fig.R3a1). Note that for this small network, $R_{\text{ego-1}}$ is the same as the cocktail R_d designed by only considering the direct inhibitors, while $R_{\text{ego-3}}$ is actually equivalent to R_{global} . Moreover, $R_{\text{ego-1}}$ and $R_{\text{ego-2}}$ just represent two subsets of R_{global} . Though the performance of $R_{\text{ego-1}}$ and $R_{\text{ego-2}}$ are not comparable with $R_{\text{ego-3}} = R_{\text{global}}$, they both can suppress the abundance of *C. difficile* to a much lower level than that of the diseased state. For comparison purposes, we also show the performance of three other cocktails (R_1 , R_2 and R_3), representing three randomly chosen subsets of R_{global} . We found that none of them is comparable with R_{global} . This result clearly demonstrates the rational design of R_{global} . We emphasize that R_{global} is designed based on the specific disrupted microbiota, hence it is “personalized”. For a different disrupted microbiota (e.g., as shown in Fig.R3c2), we can design a different R_{global} , which again outperforms any other cocktails (Fig.R3d).

Overall, this result demonstrates the necessity and advantages of considering the ecological network when designing probiotic cocktails.

2. The meaning of the universality of microbial dynamics is unclear.

By “universality of microbial dynamics” we mean that different local communities share the same underlying dynamic rules, which can be formalized in a generic population dynamics model parameterized with the same set of intrinsic species growth rates and inter-species interactions. Then, the main difference between different local communities is just the species collections they have. Specifically, for host-associated microbial communities, the assumption of universal dynamics means that species interact with each other in a host-independent way, and microbial communities associated with different hosts can just be modeled as different local communities with the same population dynamics model but different and highly personalized species collections. The presence of universal dynamics is crucial to the design of intervention strategies to manipulate microbial communities. Namely, if the dynamics universal, one can design general intervention strategies; by contrast, if the dynamics are strongly host-specific, truly personalized interventions are mandatory: a personalized intervention must consider not only the unique microbial composition of each host but also the specific dynamic rules of the underlying ecosystem.

In a previous work [R20], we developed a computational method to detect universality in microbial dynamics using the Human Microbiome Project (HMP) database. The basic idea is that the more species two microbiome samples share (i.e., higher overlap) the higher should be the similarity between their species composition (i.e., lower dissimilarity). This “fingerprint” of universal dynamics can be detected as a negative slope in a dissimilarity-overlap curve (DOC) (Fig.R5a). By contrast, if dynamics are not universal and hence host-specific, then overlap and dissimilarity should not display any relationship, rendering a flat DOC (Fig.R5b). Applying the DOC method to cross-sectional data from HMP, we found that gut microbiomes of healthy adults display pronounced universal dynamics (Fig.R5c-e). This fact motivates our assumption of universal dynamics.

In our previous work [R20], we also analyzed gut microbiome samples from 17 rCDI patients. Interestingly, the universality of their gut microbial dynamics was not observed pre-FMT but was observed post-FMT. We hypothesize that the possibly universal microbial dynamics of the rCDI subjects are undetectable by our DOC method for many reasons (for details, see main text page 7, lines 221-228). This prompts us to systematically study the impact of host-dependent microbial dynamics on the efficacy of FMT (see the main text Fig.3).

References:

- [R1] Sugihara, G. *et al.* Detecting Causality in Complex Ecosystems. *Science* **338**, 496–500 (2012).
- [R2] Fisher, C. K. & Mehta, P. Identifying Keystone Species in the Human Gut Microbiome from Metagenomic Timeseries Using Sparse Linear Regression. *PLoS ONE* **9**, e102451 (2014).
- [R3] Stein, R. R. *et al.* Ecological modeling from time-series inference: insight into dynamics and stability of intestinal microbiota. *PLoS Comput Biol* **9**, e1003388 (2013).
- [R4] Bucci, V. *et al.* MDSINE: Microbial Dynamical Systems INference Engine for microbiome time-series analyses. *Genome Biology* **17**, (2016).
- [R5] Xiao, Y. *et al.* Mapping the ecological networks of microbial communities. *Nature Communications* **8**, 2042 (2017).
- [R6] David, L. A. *et al.* Host lifestyle affects human microbiota on daily timescales. *Genome Biology* **15**, R89 (2014).
- [R7] Faith, J. J. *et al.* The Long-Term Stability of the Human Gut Microbiota. *Science* **341**, 1237439 (2013).
- [R8] Zoetendal, E. G., Akkermans, A. D. L. & De Vos, W. M. Temperature Gradient Gel Electrophoresis Analysis of 16S rRNA from Human Fecal Samples Reveals Stable and Host-Specific Communities of Active Bacteria. *Appl Environ Microbiol* **64**, 3854–3859 (1998).
- [R9] Caporaso, J. G. *et al.* Moving pictures of the human microbiome. *Genome Biology* **12**, R50 (2011).
- [R10] David, L. A. *et al.* Diet rapidly and reproducibly alters the human gut microbiome. *Nature* **505**, 559–563 (2014).
- [R11] O'Brien, C. L., Allison, G. E., Grimpen, F. & Pavli, P. Impact of Colonoscopy Bowel Preparation on Intestinal Microbiota. *PLoS One* **8**, (2013).
- [R12] Knights, D. *et al.* Rethinking “Enterotypes”. *Cell Host & Microbe* **16**, 433–437 (2014).
- [R13] Gibson, T. E. The human microbiome: Opportunities for dynamics, systems, and control (based on the IFAC blog post translate or die). in *2016 American Control Conference (ACC)* 7340–7345 (2016).
- [R14] Huttenhower, C. *et al.* Structure, function and diversity of the healthy human microbiome. *Nature* **486**, 207–214 (2012).
- [R15] Staley, C. *et al.* Predicting recurrence of *Clostridium difficile* infection following encapsulated fecal microbiota transplantation. *Microbiome* **6**, 166 (2018).
- [R16] Juul, F. E. *et al.* Fecal Microbiota Transplantation for Primary *Clostridium difficile* Infection. *N Engl J Med* **378**, 2535–2536 (2018).
- [R17] Quraishi, M. N. *et al.* Systematic review with meta-analysis: the efficacy of faecal microbiota transplantation for the treatment of recurrent and refractory *Clostridium difficile* infection. *Aliment Pharmacol Ther* **46**, 479–493 (2017).
- [R18] Allegretti, J. R. *et al.* Recurrent *Clostridium difficile* Infection Associates with Distinct Bile Acid and Microbiome Profiles. *Aliment Pharmacol Ther* **43**, 1142–1153 (2016).
- [R19] Friedman, J., Higgins, L. M. & Gore, J. Community structure follows simple assembly rules in microbial microcosms. *Nature Ecology & Evolution* **1**, 0109 (2017).
- [R20] Bashan, A. *et al.* Universality of human microbial dynamics. *Nature* **534**, 259–262 (2016).
- [R21] Faust, K. & Raes, J. Host-microbe interaction: Rules of the game for microbiota. *Nature* **534**, 182–183 (2016).

Figure R1 | Interaction matrix and its corresponding contribution matrix of two real microbial communities. **a**, The interaction network of a community composed of 8 soil bacterial species was inferred by the temporal data of solos and duos [R19]. Red edges mean the direct promotion while blue edges indicate the direct inhibition. **b**, The local interaction matrix and the corresponding contribution matrix of subcommunities consisted of 3 species. Counter-intuitive effects are indicated by red boxes. **c**, The interaction matrix of an ecological network inferred from mice experiments on antibiotic-mediated CDI [R3]. **d**, The contribution matrix of ecological network of **c**. Zero rows and columns indicate three species that get extinct in this community. The red boxes indicate the counter-intuitive effects of the remaining community.

Figure R2 | Impact of taxonomic diversity of recipients' pre-FMT microbiota on the FMT efficacy and the donor-recipient compatibility issue. **a**, In this clinical trial [R15], the fecal material of each donor was used in FMT for different recipients. For a typical donor, some recipients responded to FMT (yellow lines), some didn't (blue lines). The trial collected samples at different time points: pre-FMT, 2-6 days post-FMT, weeks (7-20 days) post-FMT, months (21-60 days) post-FMT, and long-term (> 60 days). **b**, The trajectories of recipients' samples from pre-FMT to final post-FMT are visualized in the PCoA plot (using Bray-Curtis dissimilarity between samples at genus level). Pre-FMT, post-FMT and final samples are represented as squares, small dots, and triangles, respectively, with colors indicating the different time points as shown in in panel a. The taxonomic diversity of the responders' and non-responders' pre-FMT microbiota at the OTU level are compared by using three indices: **(c)** species richness ($p = 0.18$), **(d)** Shannon diversity ($p = 0.13$), and **(e)** Simpson diversity ($p = 0.28$). Hypothesis testing for differences of the means were done by a linear mixed effects analysis using treatment as fixed effects and donor ID as a random effect. The linear mixed model was fit to data via REML (restricted maximum likelihood), using the **lme4** package in R. The p-values were computed via the Satterthwaite's method, using the **lmerTest** package in R. The black line represents mean value of the points. The shape of each data point in c-e is consistent with that of the recipient's corresponding donor as shown in panel a. **f-j**, Ternary plot of fractions of donor-specific taxa (f_d), recipient-specific taxa (f_r), and common taxa (f_c) for each (donor, recipient) pair at different time points. **f**, pre-FMT. **g**, days post-FMT. **h**, weeks post-FMT. **i**, months post-FMT. **j**, long-term post-FMT.

Figure R3 | Probiotic cocktails designed based on the ecological network and the specific disrupted microbiota can effectively decolonize *C. difficile*. **a1**, An ecological network involving the GnotoComplex microflora (a mixture of human commensal bacterial type strains) and *C. difficile* was inferred from mouse data [R4]. Node C represents *C. difficile*. The edge width and node size indicate the inter-species interaction strength and the intrinsic growth rate, respectively. Red (or blue) edges indicate the direct promotion (or inhibition), respectively. **a2**, A disrupted microbiota due to a hypothetical antibiotic administration. **a3**, The restored microbiota due to the administration of a particular probiotic cocktail. **b**, The trajectory of *C. difficile* abundance over three different time windows: (1) the initial healthy microbiota, (2) the disrupted microbiota, and (3) the microbiota post probiotic administration. In the third time window, we compare the performance of various probiotic cocktails in terms of their ability to decolonize *C. difficile*. Those cocktails were designed by considering direct inhibitors only (R_d), the global ecological network (R_{global}), the n -step ego-networks of *C. difficile* with $n = 1, 2, 3$ (R_{ego-1} , R_{ego-2} and R_{ego-3}), and randomly chosen subsets of R_{global} (R_1 , R_2 and R_3). Note that R_{ego-1} is not necessarily always the same as R_d because R_{ego-1} needs to consider the net impact among neighbors of the focal species. **c1-c3**, We start from the same initial microbiota as shown in panel a1. But another hypothetical antibiotic administration can lead to a different disrupted microbiota (c2), which can be restored through probiotic administration (c3). **d**, Performance of different probiotic cocktails in decolonizing *C. difficile* vary.

Figure R4 | Stability of human gut microbiota visualized through longitudinal data. a, Background in blue is scope of variation for the Human Microbiome Project (HMP [R14], cross sectional cohort of $n = 350$). Yellow Dots are from a single male subject, ($n = 336$ longitudinal almost daily samples), and orange dots are from a female subject, ($n = 131$ longitudinal less frequently sampled but over the same time window) and are from the Moving Pictures Study (MPS) [R9]. HMP and MPS samples were analyzed using the same OTU picking scheme, and were projected into two principal components using Bray-Curtis dissimilarity. An instance where the samples deviated from the steady region for the female (**b**) or male (**d**), but then returning to the steady region rapidly after each deviation. A typical multiple day snapshot of the female (**c**) or male (**e**) samples with all samples staying in the same region.

Figure R5 | Detecting the universality of microbial dynamics. **a**, If microbial community dynamics are universal between individuals (A–C), the presence of the same species (species represented by colored nodes; grey nodes represent absent species) should also lead to similar species proportions and a negative DOC slope. **b**, If the community dynamics are host-specific, e.g., the microbial interactions (promotive, inhibitive, or neutral) are strongly host-dependent, the presence of the same species does not lead to similar proportions and the DOC is flat. The panels **a**, **b** are adopted from [R21]. **c**, Four gut microbial sample pairs (i–iv) represented by stacked bars at the genus level. For each sample pair, their shared genera are colored while non-shared genera are shown in grey. **d**, DOC (in dark blue) of gut microbial sample pairs from the HMP study ($M = 190$ samples). Grey dots represent all the 17,955 sample pairs. **e**, DOC (in dark red) of the randomized samples is flat. In **b** and **c**, and throughout the paper, shaded area indicates the range of the 94% confidence interval. The panels **c**–**e** are adopted from [R20].

Figure R6 | Simulated time series of species abundances before and after a hypothetical antibiotic administration. The time series is the same as shown in the main text Fig.2g, but here we highlight all the 4 species (in thick lines) that are resistant to the antibiotic and display increased abundance after antibiotic administration.

Figure R7 | An ecological network of a microbial community and species' abundances evolving from an initial state to final steady state. Consider a microbial community of 15 species. The ecological network is the same with Fig.2a in the main text. Blue (or red) edges represent the inhibition (or promotion) effects between two species. If this local community evolves from an initial state, species C, 1, 7 present at the beginning become extinct when they the community reaches its steady state.

Reviewers' comments:

Reviewer #1 (Remarks to the Author):

Dear Authors, I had several concerns with this paper and I can say that you have addressed all of them in a satisfactory way. Open questions and limitations remain, of course, but these are now more like discussed than neglected, which makes the manuscript very good. These are the limitations of the scientific field, not of the paper.

Reviewer #2 (Remarks to the Author):

Liu and colleagues have done a good job of responding to the reviewer comments. However, I still have a few concerns:

1) From page 16 of rebuttal: The authors state "Figure R1a illustrates an ecological network inferred from the abundance data of a synthetic soil microbial community of eight species [R19]."

This soil data seems very out of place in this manuscript, and has nothing to do with CDI and FMT. The synthetic soil community isn't designed to inhibit a pathogen. There are defined communities that have been developed to treat rCDI - is that data available? If not, I think it's best to exclude the soil data as these communities are so different.

2) From page 17 of rebuttal: The authors state "The optimal probiotic cocktail Rglobal is designed based on the global ecological network (Fig.R3a1) and the specific disrupted microbiota (Fig.R3a2). This cocktail Rglobal contains two direct inhibitors of *C. difficile* (i.e., species-4: *C. scindens* and species-13: *R. hominis*), and two indirect inhibitors of *C. difficile* (i.e., species-5: *R. obeum* and species-12: *K. oxytoca*)."

K. oxytoca is an opportunistic pathogen, and would never be approved for use in a synthetic cocktail. Therefore, the statement that it's part of an optimal synthetic cocktail doesn't make any sense. Moreover, *Klebsiella* are often enriched in the gut microbiota of rCDI patients pre-FMT and at very low levels or completely absent post-FMT/in FMT donors. How can the authors explain this finding?

Reviewer #3 (Remarks to the Author):

I reviewed the revised version of the article “An Ecological Framework to Understand the Efficacy of Fecal Microbiota Transplantation”. Improvements were made, but the key problems of the article remain. Furthermore, the article is still much too long, and it is often unclear what the new finding of each lengthy section is.

From what I can tell, a computer simulation was developed that, starting from an initial pool of species and fixed ecological interaction terms, simulates a gLV system. Additionally, the system allows to inject other species at later times. This is used to investigate principles of FMT success or failure. Of course, as echoed by the other reviewers, failure of a species to establish after FMT in this set up is likely due to out-competition of invaders by resident strains. Yet, interactions via third species may lead to scenarios where direct competitors can help each other (or vice versa, species that benefit each other pairwise may compete via interactions with third species). I would have wished for a clear narrative that lays this out rather than a meandering story full of far reaching claims that such a ODE system will never be able to support with much meaning. The main result, as per the abstract, is now a concept to develop personalized microbial cocktails. Yet, as far as I can see, the corresponding results are from trial and error simulations that assume a completely understood microbial ecosystem. I do not see this as very useful at this point. Therefore, this paper is a missed opportunity: it would have been interesting, from a purely theoretical perspective, to better understand how ‘counter-intuitive’ interactions could manifest/be circumvented – does your theoretical work reveal any ‘tricks’ or global patterns that facilitate development of personal FMT solutions (beyond trial and error)? A majorly trimmed down, more focused version of this article might be a great improvement.

Detailed comments

Section 2.1 shows that simulations of ODEs can be used to simulate FMT. In 100 lines it is explained that Lotka Volterra ODEs were solved, with the tweak that allows introduction of antibiotics as well as species (“FMT”) at any point. This section is extremely long, including a lengthy discussion of functional response types, and yet it is missing details (including which functional response was used, and further issues below) that matter for the many claims made. What is the main result of this section?

L160: "...reaches steady state (Fig.2b). Note that some species might become extinct during the process, which just means that those N species..." As per my previous review, your ODEs do not allow for extinction. Do you implement thresholds where you set values to zero? This might often be okay, but here major perturbations to a dynamic system are under investigation. This means even tiny residual cell numbers (that were perhaps set to zero since your equations will never reach zero) may indeed come back exponentially. Because you study C diff, which forms spores, might these tiny numbers not matter? At least explain your methods and how you implement cutoffs.

Figure 2 caption, L907: "...Note that seven species can survive after the hypothetical antibiotic administration,..." Maybe 'simulated' antibiotic administration?

Section 2.2 reviews a recent publication by the authors describing evidence that interactions between microbes are host-independent ("universal"). As pointed out in my previous review, this section then engages in a tautological argument: if all interactions were host specific, then microbes interact differently in different hosts and since interactions determine the success of FMT, FMT may fail. The section concludes with the statement that, going forward, only universal interactions are considered. Since the authors themselves provided evidence for universal interactions, and correctly point out that FMT success indicates likely low import of host specificity, this section could be shrunk (two full paragraphs reviewing your own article seems a lot) to reduce overall article length.

Section 2.2.1: Can high diversity and C diff infection come together? It would be good to discuss the range of actual diversities in rCDI patients, and say whether the sharp drop off in FMT success around ~ 0.92 Simpson is meaningful.

L367: "Note that autologous FMT has been shown to be a safe and effective way to help replenish beneficial gut bacteria in cancer patients who require intense antibiotics during allogeneic hematopoietic stem cell transplantation⁹⁵. In our simulations, we find that autologous FMT will always yield high recovery degree, regardless of the taxonomic diversity in the pre-FMT microbiota (see bottom rows in Fig.5a-c)"

The very trial you are referring to is an autologous FMT, and absolutely does not always yield high recovery degrees. Maybe you should discuss the limited ability of your simulations to predict the

outcome of this trial, what would your simulations have predicted, and what was actually found in the data?

L42: “Recently, FDA issued an urgent warning¹⁵ regarding FMT, as two immunocompromised adults who received investigational FMT developed invasive infections. One of the individuals died. This tragedy underscores need for greater understanding of FMT.” This was a matter of inappropriate screening, in particular for multi resistant strains. Your research has little to no relevance to this, and this new bold claim reads a little macabre.

RESPONSE to Reviewer #1

Dear Authors, I had several concerns with this paper and I can say that you have addressed all of them in a satisfactory way. Open questions and limitations remain, of course, but these are now more like discussed than neglected, which makes the manuscript very good. These are the limitations of the scientific field, not of the paper.

We thank Reviewer #1 very much for reviewing our paper again. We are very pleased to know that s/he is now happy with the revised version. We share her/his opinion that remaining limitations are the limitations of the scientific field, rather than this particular paper.

Response to Reviewer #2

Liu and colleagues have done a good job of responding to the reviewer comments. However, I still have a few concerns:

We thank Reviewer #2 very much for reviewing our paper again. We are glad to know that s/he appreciated our responses to her/his previous comments. Next we address each of her/his remaining concerns.

Comments:

1) From page 16 of rebuttal: The authors state "Figure R1a illustrates an ecological network inferred from the abundance data of a synthetic soil microbial community of eight species [R19]."

This soil data seems very out of place in this manuscript, and has nothing to do with CDI and FMT. The synthetic soil community isn't designed to inhibit a pathogen. There are defined communities that have been developed to treat rCDI - is that data available? If not, I think it's best to exclude the soil data as these communities are so different.

We thank Reviewer #2 for pointing this out. We fully agree with her/him that the synthetic soil community seems out of place in this manuscript. We have excluded it from the revised main text. The key reason why in the previous version we presented the soil data is that we would like to show the network effect can be observed from both *in vitro* and *in vivo* microbial communities.

There are several examples of defined communities or bacterial consortium that have been developed by research labs or therapeutics companies to treat CDI in mice and humans. This includes:

- a simple mixture of six phylogenetically diverse intestinal bacteria that clear *C. difficile* 027/BI infection from mice [R23];
- a defined mixture of 33 fecal bacterial strains in the RePOOPulate study [R24];
- and VE303 (consisting of 8 types of clonal human commensal bacteria strains) developed by Vedanta Biosciences [R25].

Unfortunately, the underlying ecological networks of those defined communities are not publicly available. (For VE303, even the exact species composition is not publicly available.) Hence, it is simply impossible for us to analyze those defined communities using our framework.

2) From page 17 of rebuttal: The authors state "The optimal probiotic cocktail Rglobal is designed based on the global ecological network (Fig.R3a1) and the specific disrupted microbiota (Fig.R3a2). This cocktail Rglobal contains two direct inhibitors of *C. difficile* (i.e., species-4: *C. scindens* and species-13: *R. hominis*), and two indirect inhibitors of *C. difficile* (i.e., species-5: *R. obeum* and species-12: *K. oxytoca*)."

K. oxytoca is an opportunistic pathogen, and would never be approved for use in a synthetic cocktail. Therefore, the statement that it's part of an optimal synthetic cocktail doesn't make any sense.

We thank Reviewer #2 for this very insightful comment. In Fig.R3 of the previous response letter, the ecological network involves *C. difficile* and the so-called GnotoComplex microflora

(which contains *K. oxytoca*). Based on the original publication [R4], GnotoComplex is a set of defined human commensal bacteria, where strains were chosen to approximate phylogenetic diversity and key roles of the microbiota in the host, including the ability to transform bile acids and degrade a variety of dietary compounds.

We fully agree with Reviewer #2 that *K. oxytoca* is an opportunistic pathogen and should not be used in any synthetic cocktail to treat CDI. We have mentioned this point in the revised manuscript (see **main text, page 13, lines 468-471**). Moreover, we performed additional simulations, showing that even without *K. oxytoca*, the “near-optimal” cocktail can still decolonize *C. difficile* to a large extent (see **Fig.R8**).

Moreover, *Klebsiella* are often enriched in the gut microbiota of rCDI patients pre-FMT and at very low levels or completely absent post-FMT/in FMT donors. How can the authors explain this finding?

We thank Reviewer #2 for pointing out this very interesting phenomenon.

- *First*, using the network shown in Fig.R3, we can actually demonstrate this phenomenon numerically. As shown in **Fig.R9**, the abundance of *K. oxytoca* was very low for the initial “healthy” community. After simulated antibiotics, the abundance of *K. oxytoca* increased to a very high level. Interestingly, after introducing back those species eradicated by antibiotics (which roughly mimics the effect of FMT), the abundance of *K. oxytoca* decreased to a very low level. This example clearly demonstrates the power of our ecological modeling framework in reproducing real-world observations.
- *Second*, we think this observation implies that the abundance of *K. oxytoca* is actually co-varying or positively correlated with that of *C. difficile*. To further verify this point, we performed extensive simulations, finding that there is indeed a positive correlation between the abundance of *K. oxytoca* and that of *C. difficile* (see **Fig.R10**). Based on the ecological network shown in Fig.R3, those two species do not directly interact with each other. Hence, the positive correlation of their abundances is largely due to those “mediator” species that interact with both of them.

Finally, we thank Reviewer #2 again for carefully reviewing our manuscript. We hope our responses above have addressed her/his remaining concerns in a satisfactory manner.

Response to Reviewer #3

I reviewed the revised version of the article “An Ecological Framework to Understand the Efficacy of Fecal Microbiota Transplantation”. Improvements were made, but the key problems of the article remain.

We thank Reviewer #3 very much for reviewing our paper again. Next we address each of her/his remaining concerns.

Furthermore, the article is still much too long, and it is often unclear what the new finding of each lengthy section is.

We thank Reviewer #3 for this critical comment. Regarding the length of our manuscript, we are really sorry about that. In the first round of peer review, all reviewers raised concerns about the lack of validation of our simulation results using real data. To address this critical common concern, we had to add several new sections/figures to the revised manuscript, which unavoidably further increased the total length of the paper.

Regarding the findings presented in each section, we think the self-explained title of each section/subsection somehow summarizes its key result/finding. To demonstrate that, here we show the outline of our paper:

1. Introduction

2. Theoretical Results

2.1 Ecological Modeling Framework

2.2 Reveal Key Factors that Determine the Success of FMT

2.2.1 Impact of host-dependent microbial dynamics on FMT efficacy

2.2.2 Impact of taxonomic diversity of recipient’s pre-FMT microbiota on FMT efficacy

2.2.3 Impact of donor-recipient compatibility on FMT efficacy

2.3 Towards a rational design of probiotic cocktails to treat rCDI

3. Real Data Analysis

3.1 Network effect in real microbial communities

3.2 Taxonomic diversity of pre-FMT microbiota of responders and non-responders

3.3 The donor-recipient compatibility issue in a clinical trial

3.4 Rational design of probiotic cocktails for a real microbial community

4. Discussion

Finally, in the revised version of our manuscript we have tried our best to shorten certain parts of the paper (following the excellent suggestions made by Reviewer #3). In total, we have shortened the main text by **993 words**.

From what I can tell, a computer simulation was developed that, starting from an initial pool of species and fixed ecological interaction terms, simulates a gLV system. Additionally, the system allows to inject other species at later times. This is used to investigate principles of FMT success or failure. Of course, as echoed by the other reviewers, failure of a species to establish after FMT in this set up is likely due to out-competition of invaders by resident strains. Yet, interactions via third species may lead to scenarios where direct competitors can help each

other (or vice versa, species that benefit each other pairwise may compete via interactions with third species). I would have wished for a clear narrative that lays this out rather than a meandering story full of far reaching claims that such an ODE system will never be able to support with much meaning.

We thank Reviewer #3 for this very critical comment.

We emphasize that the phenomena described by Reviewer #3 (i.e., *direct competitors can help each other due to interactions via third species; species that benefit each other pairwise may compete via interactions with third species*) are clear demonstrations of the so-called “**network effect**” coined in our paper. In essence, the network effect means that the effective or net impact of species- j on species- i is really context dependent, i.e., it depends on the presence of other species. Specifically, as we mentioned in the previous response letter, if we compare the direct impact with the net impact of species- j on species- i , there are three cases:

- (1) **normal**: direct and net impacts share the same sign: $\text{sign}(a_{ij}) = \text{sign}(s_{ij})$;
 - (2) **bridging**: direct impact is zero while net impact is not: $a_{ij} = 0$ but $s_{ij} \neq 0$;
 - (3) **counter-intuitive**: direct and net impacts have opposite signs: $\text{sign}(a_{ij}) = -\text{sign}(s_{ij}) \neq 0$.
- Here, the direct impacts are encoded in the interaction matrix $A = (a_{ij})$, while the net impacts are encoded in the contribution matrix $S = (s_{ij})$.

Note that the two phenomena described by Reviewer #3 can be formalized as follows:

$$(3.1) \ a_{ij} < 0, a_{ji} < 0; \ s_{ij} > 0, s_{ji} > 0;$$

$$(3.2) \ a_{ij} > 0, a_{ji} > 0; \ s_{ij} < 0, s_{ji} < 0;$$

which can be considered as two special cases of the *counter-intuitive* case described above.

In the previous response letter, to directly demonstrate the presence of the network effect using real data, we analyzed two synthetic microbial communities constructed *in vitro* and *in vivo*, respectively. Moreover, in the previous version of our main text, we dedicated a whole subsection (Sec. 3.1) to describe the network effect using real data (both *in vitro* and *in vivo*).

Here, to explicitly and systematically demonstrate the network effect using our ecological modeling framework based on ODEs, we performed extensive simulations to quantify the fractions of the three main cases (i.e., *normal*, *bridging*, and *counter-intuitive*), as well as the two special cases (3.1) and (3.2), in synthetic ecological networks with GLV dynamics (see **Fig.R11**).

- First, we directly compared the direct impacts (encoded in the interaction matrix) in Fig.R11a, and the net impacts (encoded in the contribution matrix) in Fig.R11b, for a community of $N = 15$ species. Note that in the contribution matrix, there are in total 23 counter-intuitive cases (highlighted in red boxes), and 6 of them are special cases (3.1) and (3.2) (filled with stripe patterns).
- Fig.R11c systematically showed how the fractions of the three main cases change with the community size N . Interestingly, the fractions remain quite stable over increasing community sizes. Even for a community of only $N = 10$ species, we already see the bridging and counter-intuitive cases.
- Fig.R11d demonstrated the fractions of the three main cases with increasing connectance C of the microbial community. Here we find that the fractions change gradually over increasing connectance. In particular, denser networks (larger C) tend to have a higher fraction of counter-intuitive cases (shown in red) and a lower fraction of bridging cases (shown in green).

- Fig.R11e and f showed the fractions of the two special cases (3.1) and (3.2) with different N and C . We found that the two special cases are ubiquitous, especially in dense networks with large C .

Overall, these new results clearly demonstrate the network effect (including the two phenomena mentioned by Reviewer #3) is well supported by our ecological modeling framework.

The main result, as per the abstract, is now a concept to develop personalized microbial cocktails. Yet, as far as I can see, the corresponding results are from trial and error simulations that assume a completely understood microbial ecosystem. I do not see this as very useful at this point. Therefore, this paper is a missed opportunity: it would have been interesting, from a purely theoretical perspective, to better understand how ‘counter-intuitive’ interactions could manifest/be circumvented – does your theoretical work reveal any ‘tricks’ or global patterns that facilitate development of personal FMT solutions (beyond trial and error)? A majorly trimmed down, more focused version of this article might be a great improvement.

We thank Reviewer #3 for this critical comment.

We apologize for not clearly describing our algorithm of personalized cocktail design in the previous version of the manuscript. Here, we emphasize that our algorithm is not based on trial and error simulations, but based on the ecological network and systematic study of the network effect. As detailed in Supplementary Note 3.1, if we leverage the global ecological network, our algorithm consists of the following steps:

- **Step 1.** Calculate the contribution matrix S from the interaction matrix A of the metacommunity, quantifying the net impact between any two species.
- **Step 2.** Consider both direct ($a_{cj} < 0$) and effective ($s_{cj} < 0$) inhibitors as the “global inhibitors” from the metacommunity. Let the initial cocktail contain all those global inhibitors that are not present in the diseased local community (i.e., the patient’s disrupted microbiota).
- **Step 3.** If transplanting the initial cocktail to the patient microbiota will decolonize *C. difficile*, the procedure terminates. If not, go to Step 4.
- **Step 4.** Calculate the local contribution matrix using the new local community consisting of all species in the patient’s diseased microbiota and all species in the initial cocktail. For each species in the initial cocktail, we numerically test if it is an effective inhibitor (i.e., has a negative net impact on the growth of *C. difficile*) in the restored local community. The species is kept in the cocktail if it is an effective inhibitor, and it is removed from the cocktail if it is not an effective inhibitor.
- **Step 5.** Repeat Step 4 until all the species in the cocktail are effective inhibitors in the local community.
- **Step 6.** Return the final cocktail as the personalized probiotic cocktail.

To better explain the workflow of our algorithm, we also presented a schematic diagram (see **Fig.R12**) and added this diagram to **Supplementary Note 3.1**.

We admit that, at first glance, the iterative nature of our algorithm might read like a trial-and-error approach. But we hope Reviewer #3 can now appreciate that this is really not the case because we explicitly consider the network effect. In the revised main text, we further emphasized this point (see **page 10, lines 351-353**). Moreover, as we already demonstrated in

Fig.R3 of the previous response letter, the resulting optimal personalized cocktail R_{global} indeed displays the best performance in decolonizing *C. difficile*.

We agree with Reviewer #3 that the above algorithm does require “a completely understood microbial ecosystem”, i.e., the global ecological network of the microbial community (which is currently unavailable for the human gut microbiome). But in the previous response letter (Fig.R3) and main text (Figs.6 and 9), we have already presented compelling evidence that even if we don't know the global ecological network of the human gut microbiota, knowing the **ego network** of *C. difficile* can still help us design a near-optimal personalized probiotic cocktail to decolonize *C. difficile*. Here, the ego network of *C. difficile* consists of a focal species (“ego”, i.e., *C. difficile*), those species to which *C. difficile* directly interact with (they are called “alters”), the interactions between *C. difficile* and its alters, as well as the interactions among the alters. The algorithm to design a probiotic cocktail based on the ego network of *C. difficile* is actually very similar to the algorithm based on the global ecological network. The only difference is that we need to construct the initial tentative probiotic cocktail based on the ego network (see Supplementary Note 3.2 for details). Since inferring the ego network of *C. difficile* should be much easier than inferring the global ecological network of the human gut microbiota, we think our algorithm holds great promise for the rational design of personalized probiotic cocktails to treat rCDI.

Detailed comments

Section 2.1 shows that simulations of ODEs can be used to simulate FMT. In 100 lines it is explained that Lotka Volterra ODEs were solved, with the tweak that allows introduction of antibiotics as well as species (“FMT”) at any point. This section is extremely long, including a lengthy discussion of functional response types, and yet it is missing details (including which functional response was used, and further issues below) that matter for the many claims made. What is the main result of this section?

We thank Reviewer #3 for this comment. We apologize for the lengthy description of our modeling framework. The main purpose of that section is to explain the details of this modeling framework. We agree with Reviewer #3 that certain parts of this section (e.g., the discussion of different functional response types) are redundant and should be trimmed down. In the revised manuscript, we have tried our best to do that. We only keep those details that are needed for readers to understand the results presented in late sections.

Regarding which functional response was used, we apologize for not making this point explicit in the previous version of the manuscript. In all our simulations, we used the GLV model, implying that we were adopting the linear functional response. In the revised main text, we explicitly mentioned this point (**page 4, lines 129-130**):

“The GLV model has been used in several ecological modeling works of host-associated microbial communities^{67,70,71}. In this work, we also use it to simulate the FMT process.”

L160: “...reaches steady state (Fig.2b). Note that some species might become extinct during the process, which just means that those N species...” As per my previous review, your ODEs do not allow for extinction. Do you implement thresholds where you set values to zero? This might often be okay, but here major perturbations to a dynamic system are under investigation. This means even tiny residual cell numbers (that were perhaps set to zero since your equations will

never reach zero) may indeed come back exponentially. Because you study *C. difficile*, which forms spores, might these tiny numbers not matter? At least explain your methods and how you implement cutoffs.

We thank Reviewer #3 very much for this critical comment. We apologize for not fully addressing this comment in the previous response letter.

We fully agree with Reviewer #3 that, **mathematically, the structure of the ODEs in the GLV model does not allow for natural extinction in any finite time**, i.e., that a species with initial abundance $x_i(0) > 0$ reaches the value $x_i(T) = 0$ for some finite time $T < \infty$. (Note that the case of species eradication due to simulated antibiotic administration certainly doesn't count as natural extinction.) Therefore, during a natural evolution of the GLV model (without any simulated "antibiotics"), for any finite time $t > 0$, the species abundance $x_i(t)$ would never be exactly zero, unless it was absent at $t = 0$. This is because the coordinate planes ($x_i = 0$) are invariant manifolds for the GLV model, and since solutions of initial value problems for the GLV model are unique, it follows that natural extinction cannot occur in any finite time [R26]. Any trajectory of the system with positive initial position remains in the first orthant for all time.

What we intended to show in Fig.R7 is that those initially present species might not co-exist in the long run. Mathematically, this means that a certain subset of initially present species will **go to extinction asymptotically**, i.e., their abundances vanish with time: $\lim_{t \rightarrow \infty} x_i(t) = 0$. Such asymptotic extinctions in the GLV model has been heavily studied before [R26]-[R30]. Those previous studies typically focused on special interaction types (either predator-prey or competitive interactions) so that the interaction matrix A contains special sign patterns (either $\text{sign}(a_{ij}) = -\text{sign}(a_{ji})$ or $\text{sign}(a_{ij}) = \text{sign}(a_{ji}) < 0$ for all $i \neq j$), and then derived algebraic criteria on the model parameters (r, A) , which guarantee that some species are driven to extinction asymptotically. Unfortunately, up to our knowledge, for general interaction types (or arbitrary sign patterns of the interaction matrix A), there are no analytical results to predict which species will be driven to extinction asymptotically.

We emphasize that in our FMT simulations, we actually don't have to forcibly set species abundances to zero even if they will go to extinction asymptotically. We fully agree with Reviewer #3 that some tiny residual species could recover to a high level of abundance after transplantation. Indeed, as shown in **Fig.R13e** (note that Fig.R13a-d are the same as Fig.2d-g in the main text), *C. difficile* and species-1 display exponential decay in the initial healthy microbiota. Interestingly, their abundances increase to very high levels after simulated antibiotics. Finally, after the simulated FMT, their abundances display exponential decay again. This result clearly implies that tiny residual species abundances do matter in our simulations.

We admit that in our previous design of probiotic cocktails (in particular, step-3 of our algorithm), we did introduce an abundance threshold 10^{-5} to tell if the initial cocktail will lead to the (asymptotic) extinction of *C. difficile* and hence effectively decolonize *C. difficile*. Thanks to the insightful comment of Reviewer #3, we realize the limitation of this approach. **Here, to better identify those species that will go to extinction asymptotically, we propose a novel approach, which is much better than arbitrarily choosing a threshold abundance.**

(1) Theoretical Preparation.

We can prove that any asymptotical species extinction will end up with an exponential decay. Consider that species- i will go to extinction asymptotically, i.e., $\lim_{t \rightarrow \infty} x_i(t) = 0$, while all other

species will approach their equilibrium abundance $x_j^* = x_j(\infty) > 0$. Note that species- i has dynamics:

$$\dot{x}_i = x_i \left(r_i + a_{ii}x_i + \sum_{j \neq i} a_{ij}x_j \right).$$

In the limit $t \rightarrow \infty$, we are close to $x_i(t) = 0$ and $x_j(t) = x_j^*$, we can get the first-order approximation for species- i 's dynamics (by simply ignoring the second-order term $a_{ii}x_i^2$ and replacing $x_j(t)$ by x_j^*):

$$\dot{x}_i = x_i \left(r_i + \sum_{j \neq i} a_{ij}x_j^* \right) = -\lambda_i x_i$$

where we have defined a constant $\lambda_i = -(r_i + \sum_{j \neq i} a_{ij}x_j^*)$. Then it is clear that $x_i(t)$ will decay exponentially, i.e., $x_i(t) \sim e^{-\lambda_i t}$ with $\lambda_i > 0$. The above argument can be easily extended to the case of multiple species going to extinction asymptotically. Using the same simulated data (as shown in Fig.R7), but plotting the y-axis (species abundance) on the logarithmic scale, indeed we found that a few species' abundances display exponential decay, i.e., $x_i(t) \sim e^{-\lambda_i t}$ (see **Fig.R14c**), which is fundamentally different from the behavior of those co-existing species in the steady state.

(2) Numerical procedure.

First, to numerically distinguish exponential decay from steady-state behavior, we need to get each species' long-term abundance change rate λ_i , i.e., the slope of $x_i(t) \sim e^{-\lambda_i t}$ in the semi-log plot (see Fig.R14c,d), which can be obtained by fitting the asymptotical behavior of species abundances.

Second, to avoid introducing a threshold value of λ_i , we rank those species based on their λ_i values.

Third, we remove the top- K species one by one (based on the ranked λ_i values) from the system until the residual system permits a feasible equilibrium (i.e., all the residual species have positive abundance in the steady state).

- (i) In particular, each time we rearrange the species indices such that the $(N - K)$ residual species occupy the first $(N - K)$ entries, resulting in a reduced interaction matrix $A_{(K)} \in \mathbb{R}^{(N-K) \times (N-K)}$, and a reduced intrinsic growth vector $\mathbf{r}_{(K)} \in \mathbb{R}^{(N-K) \times 1}$.
- (ii) We then calculate the equilibrium of the residual system, denoted as $\mathbf{x}_{(K)}^*$, by solving the linear equations:

$$\mathbf{x}_{(K)}^* = -A_{(K)}^{-1} \cdot \mathbf{r}_{(K)}.$$

If all the $(N - K)$ residual species have positive abundances at the equilibrium $\mathbf{x}_{(K)}^*$ (corresponding to their steady-state abundances in the infinite time limit), then we conclude that the top- K species will go to extinction asymptotically.

(3) Demonstrations.

Fig.R14e demonstrates the iteration process for the synthetic community shown in Fig.R14a. Note that the initial step $K = 0$ corresponds to the original system with all the $N = 15$ species present. When solving the linear equations for equilibrium, we find negative species abundances (highlighted in blue), suggesting that the $N = 15$ species cannot co-exist in the steady state. After we remove the species with largest λ_i (i.e., species-7), the residual system still does not allow for a feasible equilibrium. Until we remove the top-5 species (i.e., species-7, 1, C, 15, 9), the resulting residual system permits a feasible equilibrium (i.e., all the residual

species have positive equilibrium abundances). We conclude that those top-5 species will go to extinction asymptotically.

Fig.R15 demonstrates the iteration process for a real microbial community associated with mouse experiments of antibiotic-mediated CDI (see main text Ref.[69] for experimental details). The interaction matrix has already been presented in main text Fig.7a. Fig.R15b showed that *Barnesiella*, und. Lachnospiraceae, and *Enterococcus* all display exponential decay in the long run of the simulation time window. Their fitted λ_i values are much larger than that of other taxa (Fig.R15c). However, the iterative process in Fig.R15d indicated that just excluding und. Lachnospiraceae and *Enterococcus* will already permit a feasible equilibrium for the residual system. Hence, we conclude that und. Lachnospiraceae and *Enterococcus* (rather than any more taxa) will go to extinction asymptotically. Note that the seemingly “large” decay rate of *Barnesiella* could be just due to the fact the simulation time is not long enough to capture the true asymptotical behavior of *Barnesiella*.

Overall, we think that the new approach for the identification of asymptotic extinction is much more robust than introducing a threshold value for the abundance change rate or the abundance itself. In the revised SI, we have explicitly mentioned this point (**see Supplementary Note 1, Remark 2**).

Figure 2 caption, L907: “...Note that seven species can survive after the hypothetical antibiotic administration,...” Maybe ‘simulated’ antibiotic administration?

Thank you for pointing this out. We have replaced “hypothetic” by “simulated”.

Section 2.2 reviews a recent publication by the authors describing evidence that interactions between microbes are host-independent (“universal”). As pointed out in my previous review, this section then engages in a tautological argument: if all interactions were host specific, then microbes interact differently in different hosts and since interactions determine the success of FMT, FMT may fail. The section concludes with the statement that, going forward, only universal interactions are considered. Since the authors themselves provided evidence for universal interactions, and correctly point out that FMT success indicates likely low import of host specificity, this section could be shrunk (two full paragraphs reviewing your own article seems a lot) to reduce overall article length.

We thank Reviewer #3 for pointing this out. We have significantly shortened that part in the revised manuscript.

Section 2.2.1: Can high diversity and C diff infection come together? It would be good to discuss the range of actual diversities in rCDI patients, and say whether the sharp drop off in FMT success around ~ 0.92 Simpson is meaningful.

We thank Reviewer #3 for this very insightful comment.

To demonstrate if “high diversity and CDI come together”, here we analyzed several CDI datasets from clinical studies. The results are shown in **Fig.R16**. We found that overall CDI patients tend to have significantly lower taxonomic diversity in their gut microbiota than that of

healthy controls, but some CDI patients do show very high taxonomic diversity in their gut microbiota.

In particular, Fig.R16a shows the distribution of Simpson diversity of pre-FMT samples for responders and non-responders, as well as their healthy donors, from the cap-FMT study analyzed in the previous response letter. We found that non-responders tend to have slightly higher Simpson index than responders. And this result is qualitatively consistent with our simulation result showing that FMT efficacy generally decreases with increasing taxonomic diversity of the pre-FMT microbiota. But as we already admitted in the previous response letter, the difference between the Simpson index of responders' and non-responders' pre-FMT microbiota is not statistically significant. This could partially be due to the imbalance between sample sizes of responders ($n = 71$) and non-responders ($n = 17$).

Besides the dataset from the cap-FMT study, we also analyzed three other CDI datasets [R31]-[R33]. Fig.R16b-d compared the Simpson index distributions of microbiome samples from healthy subjects and CDI patients. Apparently, samples from healthy subjects tend to have higher Simpson index than that of CDI patients. Unfortunately, for those CDI patients, we don't have information on their FMT treatment. Hence, we cannot directly compare Simpson index of responders and non-responders, as we did for the cap-FMT study. But still we see for some CDI patients their gut microbiota displays very high diversity.

To check if "the sharp drop off in FMT success around ~ 0.92 Simpson index" is meaningful, we performed extensive simulations with different model parameters, e.g., the community size N and network connectance C . As shown in **Fig.R17**, the recovery degree generally decreases with increasing Simpson index of the pre-FMT microbiota. But the sharp drop off of recovery degree around ~ 0.92 Simpson index (as observed in Fig.4f) is really not a very representative phenomenon. Instead, it is quite sensitive to the detailed model parameters, such as N and C .

L367: "Note that autologous FMT has been shown to be a safe and effective way to help replenish beneficial gut bacteria in cancer patients who require intense antibiotics during allogeneic hematopoietic stem cell transplantation⁹⁵. In our simulations, we find that autologous FMT will always yield high recovery degree, regardless of the taxonomic diversity in the pre-FMT microbiota (see bottom rows in Fig.5a-c)"

The very trial you are referring to is an autologous FMT, and absolutely does not always yield high recovery degrees. Maybe you should discuss the limited ability of your simulations to predict the outcome of this trial, what would your simulations have predicted, and what was actually found in the data?

We thank Reviewer #3 for this very insightful comment. We fully agree with Reviewer #3 that for those allo-HSCT patients autologous FMT does not always yield high recovery degrees of their gut microbiome (as clearly demonstrated in Fig.3b of Ref. [84]). Autologous FMT yields better recovery than the control case (i.e., no intervention) only in a statistical sense.

We emphasize that there could be many reasons contributing to the imperfect recovery of an allo-HSCT patient's gut microbiota after autologous FMT.

- *First*, we notice that various antibiotics were given to those patients throughout the study period (even after autologous FMT) for prophylactic and treatment purposes (as shown in Fig.2a of Ref. [84]).

- *Second*, dietary intake of those patients could also rapidly affect their gut microbiota compositions after autologous FMT.
- *Third*, growth factors (e.g., granulocyte colony-stimulating factor, which is typically administered to allo-HSCT patients to enhance engraftment) can also affect their gut microbiota.

All those factors were not considered in our simulations of autologous FMT. Our current modeling framework can certainly simulate the impact of antibiotics after FMT, but we cannot simulate the impact of different dietary intake or drugs on the microbial composition (which is certainly a big limitation of our current modeling framework). We have admitted this point explicitly in the revised main text (see **page 15, lines 524-525**):

“Fourth, the current modeling framework does not take into account the impact of dietary intake and drugs on the host’s microbial composition.”

L42: “Recently, FDA issued an urgent warning¹⁵ regarding FMT, as two immunocompromised adults who received investigational FMT developed invasive infections. One of the individuals died. This tragedy underscores need for greater understanding of FMT.” This was a matter of inappropriate screening, in particular for multi resistant strains. Your research has little to no relevance to this, and this new bold claim reads a little macabre.

We thank Reviewer #3 for this critical comment. We have removed this sentence from the main text.

Finally, we thank Reviewer #3 again for her/his very constructive comments and suggestions. We hope our responses above have addressed those very legitimate issues/concerns in a satisfactory manner.

References

- [R23] Lawley, T. D. *et al.* Targeted Restoration of the Intestinal Microbiota with a Simple, Defined Bacteriotherapy Resolves Relapsing *Clostridium difficile* Disease in Mice. *PLOS Pathogens* **8**, e1002995 (2012).
- [R24] Petrof, E. O. *et al.* Stool substitute transplant therapy for the eradication of *Clostridium difficile* infection: 'RePOOPulating' the gut. *Microbiome* **1**, 3 (2013).
- [R25] VEDANTA BIOSCIENCES VE303, <https://www.vedantabio.com/pipeline/ve303>
- [R26] Hallam, T. G., Svoboda, L. J. & Gard, T. C. Persistence and extinction in three species Lotka-Volterra competitive systems. *Mathematical Biosciences* **46**, 117–124 (1979).
- [R27] Ahmad, S. & Lazer, A. C. One species extinction in an autonomous competition model. in *World Congress of Nonlinear Analysts '92* (ed. Lakshmikantham, V.) (DE GRUYTER, 1996). doi:[10.1515/9783110883237.359](https://doi.org/10.1515/9783110883237.359).
- [R28] Zeeman, M. L. Extinction in Competitive Lotka-Volterra Systems. *Proceedings of the American Mathematical Society* **123**, 87–96 (1995).
- [R29] Deoca, F. M. & Zeeman, M. L. Balancing Survival and Extinction in Nonautonomous Competitive Lotka-Volterra Systems. *Journal of Mathematical Analysis and Applications* **192**, 360–370 (1995).
- [R30] Ackleh, A. S., Marshall, D. F. & Heatherly, H. E. Extinction in a generalized Lotka-Volterra predator-prey model. *Journal of Applied Mathematics and Stochastic Analysis* **13**, 287-297 (2000).
- [R31] Khanna, S. *et al.* Changes in microbial ecology after fecal microbiota transplantation for recurrent *C. difficile* infection affected by underlying inflammatory bowel disease. *Microbiome* **5**, (2017).
- [R32] Schubert, A. M. *et al.* Microbiome Data Distinguish Patients with *Clostridium difficile* Infection and Non-*C. difficile*-Associated Diarrhea from Healthy Controls. *mBio* **5**, e01021-14 (2014).
- [R33] Vincent, C. *et al.* Reductions in intestinal Clostridiales precede the development of nosocomial *Clostridium difficile* infection. *Microbiome* **1**, 18 (2013).

Figure R8 | Probiotic cocktails designed based on the ecological network and the specific disrupted microbiota can effectively decolonize *C. difficile*. **a1**, An ecological network involving the GnotoComplex microflora (a mixture of human commensal bacterial type strains) and *C. difficile* is the same with Fig.9 in the main text. Node C represents *C. difficile*. The edge width and node size indicate the inter-species interaction strength and the intrinsic growth rate, respectively. Red (or blue) edges indicate the direct promotion (or inhibition), respectively. **a2**, A disrupted microbiota due to a hypothetical antibiotic administration. **a3**, The restored microbiota due to the administration of a particular probiotic cocktail $R_{\text{near-optimal}}$. **b**, The trajectory of *C. difficile* abundance over three different time windows: (1) the initial healthy microbiota, (2) the disrupted microbiota, and (3) the microbiota post probiotic administration. In terms of the designed cocktails' ability to decolonize *C. difficile*, the third time window showed the performance of two probiotic cocktails: the optimal cocktail yielded by the global ecological network (termed R_{global}), and the near-optimal cocktail (termed $R_{\text{near-optimal}}$), which excludes the opportunistic pathogen (species-12: *K. oxytoca*) from the optimal solution.

Figure R9 | Abundance of *C. difficile* and *K. oxytoca* in the initial healthy microbiota, the disrupted microbiota, and post probiotic administration. **a1**, An ecological network is the same with Fig.9 in the main text. **a2**, A disrupted microbiota due to a hypothetical antibiotic administration is the same with Fig.9c2. **a3**, The restored microbiota is introduced all the removed species by previous antibiotic administration. **b**, The trajectory of *C. difficile* and *K. oxytoca* abundance over the initial healthy microbiota, the disrupted microbiota, and the microbiota post probiotic administration. We assume that the restored microbiota is introduced all the removed species by previous antibiotic administration.

Figure R10 | The abundances of *C. difficile* and *K. oxytoca* display positive correlation. a. The simulated temporal species abundances of the GnotoComplex microflora with the initial condition that all species are present with randomly assigned initial abundances. **b.** Spearman correlation of species abundances. Here, we generated 100 steady state samples starting from different initial species collections (randomly selected from the GnotoComplex and with randomly assigned initial abundances). The black box highlights the positive correlation between *C. difficile* and *K. oxytoca*.

Figure R11 | The ubiquity of network effect in the GLV model. **a.** The interaction matrix of a community with 15 species governed by GLV dynamics. **b.** The contribution matrix of the community in **a**. Red boxes highlighted the counter-intuitive cases, while red boxes with stripe patterns indicated the two subcases (3.1) and (3.2). **c.** Fractions of the three main cases of network effect as functions of the community size. We set the intra-species interaction $a_{ii} = -1$, the inter-species interaction $a_{ij} \sim \mathcal{N}(0, 0.2^2)$, the connectance of ecological network $C = 0.8$ (the probability that species- i interacts with species- j), the growth rate of each species $r_i \sim \mathcal{U}[0, 1]$. For each N , we ran 20 different realizations. **d.** Fractions of the three main cases of network effect

as functions of the network connectance. We set $N = 100$, $a_{ii} = -1$, $a_{ij} \sim \mathcal{N}(0, 0.2^2)$, $r_i \sim \mathcal{U}[0, 1]$. For each C , we ran 20 different realizations. **e, f.** Fraction of counter-intuitive cases of network effect as a function of community size or network connectance. Here, each bar represents the total fraction of counter-intuitive cases, and the parts with filled-in stripe patterns indicate the fractions of the two special cases (3.1) and (3.2).

Figure R12 | The workflow of our algorithm for the design of personalized probiotic cocktail.

Figure R13 | Temporal behavior of species abundances in the simulated FMT process. We start from an initially “healthy” community (a), simulate the impact of antibiotic administration by eradicating some species from the community (b), restore the healthy community by transplanting species from another healthy community (c). The simulation details of the FMT process is the same as shown in Fig.2d-g of the main text. The simulated time series of species abundances with linear (d) or logarithmic scale (e).

e

$K=0$	1	2	3	4	5	6	7	8	9	10	11	12	13	14	15
1	-0.57	2.41	-0.05	0.66	0.08	-0.28	-1.62	0.54	-0.07	-0.06	1.65	0.49	-0.03	1.58	-0.33
2	-0.45	1.43	0.13	-0.3	-0.14	0.39	X	0.11	-0.08	0.08	1.81	0.79	0.22	0.94	-0.11
3	X	1.34	0.11	-0.36	0.04	0.36	X	0.11	-0.04	0.02	1.8	0.8	0.31	0.71	-0.04
4	X	1.22	0.06	X	0.17	0.36	X	0.25	-0.03	0.02	1.67	0.66	0.31	0.71	-0.04
5	X	1.21	0.06	X	0.18	0.36	X	0.25	-0.03	0.02	1.66	0.65	0.31	0.71	X
6	X	1.22	0.06	X	0.18	0.35	X	0.25	X	0.02	1.67	0.65	0.32	0.7	X

Figure R14 | Asymptotic extinction in the generalized Lotka-Volterra model. **a.** The ecological network of a microbial community with 15 species. The ecological network is the same as shown in Fig.2a of the main text. Blue (or red) edges represent the inhibition (or promotion) impacts between species. **b.** Starting from an arbitrary initial condition, we can numerically solve the ODEs and obtain the time series of species abundances. **c.** Plotting the species abundances on the logarithmic scale demonstrates that some species' abundances decay exponentially in the long run, i.e., $x_i(t) \sim e^{-\lambda_i t}$ for large t . **d.** The decay rate λ_i obtained by fitting the asymptotic behavior of each species' abundance time series. We notice that species-1, C, 7, 9, and 15 display noticeable decay rate, suggesting that they will go to extinction asymptotically. **e.** To avoid choosing a subjective threshold value of λ_i to identify those species that will go to extinction asymptotically, we develop a heuristic method. In particular, we rank those species based on their λ_i values. Then we remove the top- K species one by one from the system until we find the residual system permits a feasible equilibrium (i.e., all the residual species have positive abundances in equilibrium). Here, the K -th row represents the equilibrium abundance profile calculated by solving the linear equation $\mathbf{x}_{(K)}^* = -A_{(K)}^{-1} \cdot \mathbf{r}_{(K)}$ for the residual system. Removed species are highlighted by 'X', residual species with negative equilibrium abundances are highlighted in blue.

Figure R15 | Asymptotic extinction in a real microbial community. The interaction matrix (inferred from mouse experiments of antibiotic-mediated CDI, see Ref.[69] of the main text) was presented in Fig.7a of the main text. **a.** Starting from an arbitrary initial condition, we can numerically solve the ODEs and obtain the time series of taxa abundances. **b.** Plotting the taxa abundances on the logarithmic scale demonstrates that the abundances of *Barnesiella*, und. Lachnospiraceae, *Enterococcus* decay exponentially i.e., $x_i(t) \sim e^{-\lambda_i t}$ for large t . **c.** The decay rate λ_i obtained by fitting the asymptotic behavior of each taxon's abundance time series. *Barnesiella*, und. Lachnospiraceae, *Enterococcus* display noticeable non-zero decay rates. **d.** The iterative process showed that excluding und. Lachnospiraceae and *Enterococcus* can already permit the residual system to have a feasible equilibrium. This suggests that und. Lachnospiraceae and *Enterococcus* will go to extinction asymptotically.

Figure R16 | Comparing the alpha diversity of the gut microbiota of CDI patients and healthy subjects. **a.** The dataset termed cap-FMT has been described in the main text. We analyzed the distribution of Simpson index for responders and non-responders' pre-FMT samples. **b.** The dataset investigated the changes in gut microbiota following FMT in 38 CDI patients [R31]. **c.** The dataset collected 338 individuals including health control, patients with CDI and non-*C. difficile*-associated diarrhea [R32]. We compared the Simpson index between healthy and CDI patients' samples. **d.** The dataset collected 50 individuals including healthy controls and hospitalized CDI patients [R33]. The Simpson index was calculated at OTU level. The p-value is calculated from t-test.

Figure R17 | The FMT efficacy is strongly affected by the taxonomic diversity of the recipient's pre-FMT microbiota (quantified by Shannon diversity). From top to bottom, the community size $N=100$, 150, and 200. From left to right, the network connectance $C=0.3$, 0.4, and 0.5. We set the intra-species interaction strengths $a_{ii} = -2$, the inter-species interaction strengths $a_{ij} \sim \mathcal{N}(0,0.2^2)$, the growth rate of each species $r_i \sim \mathcal{U}[0,1]$. We performed nonparametric regression and bootstrap sampling to calculate the trend (black line) and its 94% confidence interval (gray shadow). For each point, we simulated 20 different FMTs.

REVIEWERS' COMMENTS:

Reviewer #2 (Remarks to the Author):

I think the authors did a good job responding to the reviewer comments.

My only additional comment is that I don't like the phrase "real world data" or "real microbial communities" - perhaps this could be reworded?

Reviewer #4 (Remarks to the Author):

I think the authors have answered positively to the comments raised by the Reviewers and improved their manuscript in this revised version.

However, I still have a concern about the "Data availability" section. I was not able to find in any parts of the main text or supplementary material how to access the data used in this manuscript. This should be detailed carefully in order to guarantee reproducibility of the work.

RESPONSE to Reviewer #2

I think the authors did a good job responding to the reviewer comments.

We thank Reviewer #2 again for reviewing our paper. We are very pleased to know that s/he is now happy with the revised version.

My only additional comment is that I don't like the phrase "real world data" or "real microbial communities" - perhaps this could be reworded?

OK. In the revised version, we have rephrased those terms appropriately.

Response to Reviewer #4

I think the authors have answered positively to the comments raised by the Reviewers and improved their manuscript in this revised version.

We thank Reviewer #4 very much for reviewing our paper. We are very pleased to know that s/he appreciated our efforts in addressing the previous reviewer comments.

However, I still have a concern about the "Data availability" section. I was not able to find in any parts of the main text or supplementary material how to access the data used in this manuscript. This should be detailed carefully in order to guarantee reproducibility of the work.

We thank Reviewer #4 for this critical comment. The following "Data availability" statement has been added to the main text:

Data analyzed in this work are available at
https://github.com/xiaoyandong08/FMT_simulation_framework